# Learning Mixed Multinomial Logits with Provable Guarantees

**Yiqun Hu**
MIT
huyiqun@mit.edu

**David Simchi-Levi**
MIT
dslevi@mit.edu

**Zhenzhen Yan**
Nanyang Technological University
yanzz@ntu.edu.sg

## Abstract

A mixture of multinomial logits (MMNL) generalizes the single logit model, which is commonly used in predicting the probabilities of different outcomes. While extensive algorithms have been developed in the literature to learn MMNL models, theoretical results are limited. Built on the Frank-Wolfe (FW) method, we propose a new algorithm that learns both mixture weights and component-specific logit parameters with provable convergence guarantees for an arbitrary number of mixtures. Our algorithm utilizes historical choice data to generate a set of candidate choice probability vectors, each being close to the ground truth with a high probability. We further provide a sample complexity analysis to show that only a polynomial number of samples is required to secure the performance guarantee of our algorithm. Finally, we conduct simulation studies to evaluate the performance and demonstrate how to apply our algorithm to real-world applications.

## 1 Introduction

Multinomial logit models (MNL) are widely used in a variety of settings to predict the probabilities of different outcomes. It is termed as multinomial logistic regression in statistics and its formula for choosing the $j$-th option, $\dfrac{\exp(v_j)}{\sum_i \exp(v_i)}$, is well known as the `softmax` function for multi-class classification in machine learning. In choice modeling, it is a type of discrete choice model used to analyze and predict people's decisions.

Mixed multinomial logit (MMNL) is a mixture model consisted of multiple individual MNL components and can be adopted when the target data is heterogeneous. For instance, Yang et al. [2018] demonstrates that using a single `softmax` does not have enough capacity to model natural language and claims this as a bottleneck. To alleviate this issue, they proposed a new method called *Mixture of Softmaxes* (MoS), which has higher expressiveness and can better incorporate the contextual information. Later, new techniques are proposed on top of this architecture to further boost the efficiency, such as the Mixtape from Yang et al. [2019]. On the other hand, McFadden and Train [2000] shows that MMNL models can lift the restrictive "independence of irrelevant alternatives" (IIA) assumption present in single MNL models as well as approximate any random utility-based discrete choice models to arbitrary degree of accuracy under appropriate assumptions.

Despite its importance, the study on the learning theory of MMNL is limited. A common approach in the literature is to assume a parametric family of the mixing distribution (cf.Train [2009] and Berry et al. [1995]). But this approach suffers from the model misspecification issue, i.e., there could be systematical errors in the estimation if the assumed parametric family is different from the true one. Expectation-Maximization (EM) algorithm is a widely used non-parametric method for mixture models. But it does not guarantee the estimators' convergence to the true value of the parameters. Recent inspiring work by Jagabathula et al. [2020] creatively applies the Frank-Wolfe (FW) algorithms to estimate the MMNL model and establish the sub-linear convergence of the

proposed method. However, the major issue with their approach is that the convergence result only applies to the aggregate predictors, and cannot recover each individual mixture. In this paper, we propose a new *Stochastic Subregion Frank-Wolfe* (SSRFW) algorithm, which can generate each individual mixture within an $\epsilon$-ball to one of the ground truth vectors with a high probability.

We summarize the main contributions of this paper as follows. First, we provide a learning algorithm for MMNL models that comes with provable convergence guarantees of the estimators. In particular, each generated mixture from our algorithm is shown to be within an $\epsilon$-ball to one of the ground truth with a high probability. To the best of our knowledge, this is the first paper to prove these results for MMNL models with an arbitrary number of mixture types and minimal assumptions on the model parameters. Second, we further quantify the sample complexity of the proposed algorithm. We show that only a polynomial number of samples is required to achieve a high probability convergence guarantee.

## 1.1 Preliminaries

Consider a setting where the decision makers need to choose among a set of $[M] = \{1, \dots, M\}$ items, where each alternative has a feature vector $\boldsymbol{z}_j \in \mathbb{R}^d$. Furthermore, we assume the population consists of $K$ different components (types), each associated with a unique "preference vector" $\boldsymbol{\beta}_k \in \mathbb{R}^d$ and a mixture weight $\alpha_k$, where $\sum_{k=1}^{K} \alpha_k = 1$. In other words, $\boldsymbol{\beta}_k$'s are not drawn from a parametric mixing distribution and we consider MMNL in this paper as a type of latent class models, which is why we claim our method as *non-parametric*. This is consistent with the definitions used in other literature such as Train [2009] and Jagabathula et al. [2020].

$\boldsymbol{\beta}_k$'s can also be used as user embeddings with high interpretability for decision makers who belong to the same type, which represent the corresponding type's taste on different aspects of the alternatives.

Under linear utility assumption, the probability of choosing $j$-th item for a particular mixture $k$ is

$$q_j(\boldsymbol{\beta}_k) = \frac{\exp(\boldsymbol{\beta}_k^\top \boldsymbol{z}_j)}{\sum_{i=1}^{M} \exp(\boldsymbol{\beta}_k^\top \boldsymbol{z}_i)}$$

and the aggregated counterpart for the entire population is $\mathrm{g}_j = \sum_{k=1}^{K} \alpha_k q_j(\boldsymbol{\beta}_k), j \in [M]$.

To simplify notation, we use $\boldsymbol{q}_k$ to denote the logit vector $\boldsymbol{q}(\boldsymbol{\beta}_k) = [q_1(\boldsymbol{\beta}_k), \cdots, q_M(\boldsymbol{\beta}_k)]^\top \in \mathbb{R}^M$, and $q_{kj}$ for $\boldsymbol{q}_j(\boldsymbol{\beta}_k)$, $j \in [M]$, $\forall k$. We will also refer to these logit vectors as *choice probability vectors*.

## 1.2 Background

### 1.2.1 Related Work

Learning MMNL models remains an open-research problem. One common approach in the literature is to presume a parametric family of distributions on the parameters and apply parametric estimation methods such as the maximum likelihood estimation (MLE) method (c.f. Train [2009]) or the least square regression model to compute the parameters. A well-known work in this regime is by Berry et al. [1995]. They assume the component-specific parameters are normal distributed and proposed a two-step estimation method to learn the parameters using an aggregated market share data. A potential issue for this stream of methods is model misspecification, which leads to inaccurate predictions. In particular, if the assumed parametric family is different from the true one, there could be systematical errors in the estimation.

Non-parametric estimation is also widely recognized when learning MMNL models, with the most commonly known and used method being the Expectation-Maximization (EM) algorithm (c.f. Demp-ster et al. [1977]). Train [2008] studied three different types of EM algorithms using historical choice data from each individual decision maker. The two main restrictions for the EM algorithm are 1) the algorithm can get stuck in local optimum (hence no guarantee on the convergence of the estimators) and 2) the number of mixture types needs to be pre-specified (additional heuristics are needed if such information is not known in advance, which may also introduce undesired noises).

Jagabathula et al. [2020] recently developed an estimation method for mixture models also from a non-parametric perspective based on the Frank-Wolfe (FW) algorithm (cf. [Frank and Wolfe, 1956], Jaggi [2013], [Jaggi and Sulovskỳ, 2010], [Bach, 2013], Harchaoui et al. [2015],Lacoste-Julien and Jaggi [2015]), which is an iterative method originally designed to solve constrained quadratic optimization. They manage to establish a sublinear convergence rate of the proposed estimation method for MMNL models. But the convergence result only applies to the aggregate choice probabilities. In contrast, we provide the convergence guarantee for each individual choice probability (more details about their approach and the comparisons will be discussed in Section 2).

There is also another stream of work related to MMNL models from a theoretical learning perspective. The key idea is to solve a system of equations in terms of all the parameters and prove that there is one and only one solution (identifiability). Due to the entanglement of the individual logit parameters and mixture weights, as well as the high non-linearity in the system of equations, only 2-MNL models (MMNL with two mixture types) have been studied so far. Chierichetti et al. [2018] considers the setting where the mixture weights are equal (i.e., each mixture represents 50% of the population) and Tang [2020] recently studied 2-MNL with unknown weights. This paper enriches the theoretical learning literature for MMNL models by generalizing the analysis to an arbitrary number of mixture types and with minimal assumptions on the model parameters.

### 1.2.2 Ranking Models

Plackett-Luck (PL) models is another class of model commonly used for learning preferences. It is closely related to MNL models, with the main difference in that it relies on ranking (ordinal) data.

Let $\mathcal{X} = \{x_1, \ldots, x_m\}$ be a set of $m$ alternatives and denote $\mathcal{L}(\mathcal{X})$ as the set of linear orders (full rankings) over $\mathcal{X}$. A *ranking* $R \in \mathcal{L}(\mathcal{X})$ is $x_{i_1} \succ x_{i_2} \succ \ldots \succ x_{i_m}$ where $x_{i_1}$ and $x_{i_m}$ are the most and least preferred alternative respectively. Ranking data are obtained by repeatedly selecting items after removing the previously selected items, according to the MNL choice model. Developed by Plackett [1968, 1975], the *ranking distribution* for $R = [x_{i_1} \succ x_{i_2} \succ \ldots \succ x_{i_m}]$ is characterized by

$$\mathbb{P}(R; \beta) = \prod_{p=1}^{m-1} \mathbb{P}(x_{i_p} | \mathcal{X} \setminus \cup_{q=1}^{p-1} x_{i_q}; \beta)$$

$$= \prod_{p=1}^{m} \frac{\exp(\boldsymbol{\beta}^\top \boldsymbol{z}_{i_p})}{\sum_{q=p}^{m} \exp(\boldsymbol{\beta}^\top \boldsymbol{z}_{i_q})}$$

Similar to MMNL, we can write a mixed PL model as $\mathbb{P}(R; \boldsymbol{\alpha}, \boldsymbol{\beta}) = \sum_{k=1}^{K} \alpha_k \prod_{p=1}^{m} \frac{\exp(\boldsymbol{\beta}_k^\top \boldsymbol{z}_{i_p})}{\sum_{q=p}^{m} \exp(\boldsymbol{\beta}_k^\top \boldsymbol{z}_{i_q})}$.

Despite the same parameter representation, past work focusing on learning the parameters of general class of K-PL models usually assumes the availability of ranking data or partial preference data, also known as the *top-l* order data, which is the linear order of the top $l$ alternatives from the choice set [Oh et al., 2015, Liu et al., 2019, Zhao and Xia, 2019]. This is in general very hard to acquire in many real-world applications. In this paper, we focus on learning MMNL model using repetitive choice data, which only assumes that we have records of which single alternative the decision maker chooses from the choice set. Such data is more widely available, which enables us to conduct a real-world case study with our algorithm (see Appendix B). In addition to learning algorithms, many works focus on characterizing the identifiable conditions under different assumptions for mixed MNL or PL models [Zhao et al., 2016, Zhang et al., 2022, Zhao et al., 2022]

### 1.2.3 Motivation

While in some cases, it suffices to learn only the aggregated MMNL model, there are many situations where we want to obtain accurate estimation of the parameters for each MNL component. For instance, many important applications in operations research can benefit from knowing such individual mixture choice preferences, from minimizing the risk of model misspecification to offering personalized service such as product recommendation. Its capability of capturing the heterogeneity in the population makes it a fundamental tool for demand prediction in revenue management and supply chain management. An important application that benefits from MMNL demand models is the *multi-product pricing problem* (see Appendix C for a concrete problem formulation).

Each individual preference vectors $\boldsymbol{\beta}_k$ can be thought as a representation of the taste over different attributes with respect to different consumer types, which can be utilized by manufacturers and retailers to better design their product design and marketing strategies. As an example, Kamakura and Russell [1989] used MMNL to model brand preferences and created market structure that links the pattern of brand switching with price elasticities. The result provides a 'managerially useful description of brand competition", which then in turn allows them to explore the characteristics of competition between national brands and private labels.

From another point of view, we can also think of $\boldsymbol{\beta}_k$'s as interpretable *user embeddings*. The concept of embedding is widely used in a variety of machine learning settings, including natural language processing and computer vision, which is a relatively low-dimensional vector that can be used to quantify similarities and distances between complex and/or non-numerical objects.

## 1.3  Main Results

The main contribution of this paper is to provide a learning algorithm for MMNL models with provable convergence guarantees of the estimators. Specifically we will show that each of the generated choice probability vectors from our algorithm is within an $\epsilon$-ball to one of the ground truth vectors with a high probability. Subsequently, we provide the sample complexity analysis of the algorithm. We name our algorithm the Stochastic Subregion Frank-Wolfe (SSRFW) algorithm, while is inspired by the Frank-Wolfe framework by Jagabathula et al. [2020] but has a carefully constructed stochastic feasible region that is learned from the data.

The SSRFW framework assumes to work with repeated choice data, such as consumer panel or survey data. This is a reasonable assumption in many real-world scenarios (cf. Revelt and Train [1998] and Brownstone et al. [2000]). We assume the data consists of historical choice decisions from the target population of size $N$. Specifically, for each time period $t = 1, \ldots, T$, we assume each decision maker $i$'s choice $X_i^{(t)}$ are i.i.d categorical random variables with support $x \in [M]$ and probability mass function (PMF) specified by $\boldsymbol{q}_k$ if he/she is of type $k$.

For the MMNL to be learnable, we need to assume that none of the individual MNL is identical to the rest. Let $F$ and $G$ be the cumulative distribution functions (CDF) for two different choice probability distribution $\boldsymbol{q}_k$ and $\boldsymbol{q}_{k'}$ respectively. Define the Kolmogorov-Smirnov (KS) distance between the two distributions as $D_{\text{KS}}(\boldsymbol{q}_k, \boldsymbol{q}_{k'}) = \sup_x |F(x) - G(x)|$. Definition 1 below is adapted from a similar assumption introduced in Kalai et al. [2010] to impose some regularity in the MMNL model.

**Definition 1.** *We call a mixed multinomial logit model (MMNL) $\mathbf{g}$ $\epsilon$-standard if for any given tolerance $\epsilon$*

1. $\min_k \alpha_k \geq \epsilon$
2. $\beta_{kd} \leq \frac{1}{\epsilon}, \forall k, \forall d$
3. $D_{KS}(q_k, q_{k'}) \geq \epsilon, \forall k \neq k'$

Definition 1 imposes the following properties on the MMNL model: 1) none of the mixtures has zero weight; 2) all parameters in the model are bounded; and 3) each individual logit model has some level of separation from others. We consider this as a very mild assumption in our setting, as later we will see that we will not predetermine the number of mixtures ($K$) and if a mixture/component is not present in the observed data (since it has zero weight), the algorithm will simply not learn it. This is also one of the advantages of our algorithm over the traditional approaches, such as EM.

The main result is as follows:

**Theorem 1.** *Let $\mathbf{g} = \sum_{k=1}^{K} \alpha_k \boldsymbol{q}_k$ be a mixed multinomial logit (MMNL) model over a set of $M$ items. Assume $M \geq K$. For any $\epsilon > 0$, $0 < \delta < 1$, Algorithm 1 outputs an MMNL $\hat{\mathbf{g}} = \sum_{k=1}^{K'} \hat{\alpha}_k \hat{\boldsymbol{q}}_k$ where $K' \geq K$ such that, with probability $\geq 1 - \delta$, there exists a many-to-one mapping $\pi : k' \mapsto k, k' \in [K'], k \in [K]$ such that*

$$\left\| \hat{\boldsymbol{q}}_{k'} - \boldsymbol{q}_{\pi(k')} \right\| < \epsilon, \forall\, k',$$

*and*

$$\left| \sum_{k' : \pi(k') = k} \hat{\alpha}_{k'} - \alpha_k \right| < \epsilon, \forall\, k.$$

*where $\|\cdot\|$ is the Euclidean norm. The number of samples required by Algorithm 1 is $\mathcal{O}(\frac{1}{\epsilon^2}\log(\frac{1}{\delta}))$.*

The outline of the paper is as follows: Section 2 and Section 3 will introduce the main algorithms, namely the Stochastic Subregion Frank-Wolfe (SSRFW) and the $\mathcal{Q}$-construction algorithm, where the output of the latter serves as an input to the former. In Section 4, we will discuss the theoretical properties of the SSRFW algorithm. We conduct simulation studies in Section 5. All the detailed proofs and parts of simulation studies are included in the appendix.

## 2 Stochastic Subregion Frank-Wolfe Algorithm

Recall that we observe each decision maker $i$'s choice at time $t$: $x_i^{(t)}$ (as realization of $X_i^{(t)}$). Let $Y_i^t = [0, \cdots, 1, \cdots, 0]^\top \in \mathbb{R}^M$, where $Y_{ij}^t = 1$ if $x_i^{(t)} = j$. The observed aggregated choice probability vector for period $t$ can be expressed as

$$\mathbf{y}^t = \frac{1}{N} \sum_{i=1}^N Y_i^t$$

where $\mathbf{y}^t \in \mathbb{R}^M$. Define $\mathcal{P} = \{\boldsymbol{q}(\beta) \in \mathbb{R}^M | \boldsymbol{\beta} \in \mathbb{R}^d\}$, the set of all valid logit vectors for the given choice set and their features. Denote $\overline{\mathcal{P}}$ as its closure. Since $\mathbf{g} = \sum_{k=1}^K \alpha_k \boldsymbol{q}_k$, by definition we have $\mathbf{g} \in \text{Conv}(\overline{\mathcal{P}})$.

Our main learning objective is to minimize the difference between the observed (from data) and theoretical aggregated choice probability, as follows

$$\min_{\mathbf{g}\in\text{Conv}(\overline{\mathcal{P}})} \mathcal{L}(\mathbf{g}) \equiv \min_{\mathbf{g}\in\text{Conv}(\overline{\mathcal{P}})} \frac{1}{2} \sum_{t=1}^T \left\| \mathbf{g} - \mathbf{y}^t \right\|^2 \tag{1}$$

Note that problem (1) is a non-convex problem in $\alpha_k$ and $\boldsymbol{\beta}_k$ and is in general hard to solve. We adopt the conditional gradient approach proposed by Jagabathula et al. [2020], which we will refer to as the *original FW* algorithm. Each iteration $k$ is consisted of two steps: 1) the *supporting finding step* which searches for a new direction represented by a choice probability vector $\boldsymbol{q}^{(k)}$ (as a new latent class $k$) via solving a linear optimization subproblem in the search space $\text{Conv}(\overline{\mathcal{P}})$, and 2) the *proportion update step* which updates the mixture proportion $\alpha_i, i = 0, \ldots, k$, assigned to all choice probability vectors obtained so far. The final learning outcome $\hat{\mathbf{g}}$ is $\sum_{i=0}^k \alpha_k \boldsymbol{q}^{(k)}$ where $\boldsymbol{q}^{(0)} = \mathbf{g}^{(0)} \in \overline{\mathcal{P}}$ is a random initialization. This setup corresponds to a fully-corrective variant of the generic Frank-Wolfe algorithm, and is guaranteed to find the optimal solution to the optimization problem [Jaggi, 2013] for the decision variable $\mathbf{g}$.

The problem with the original FW approach is that the feasible region $\text{Conv}(\overline{\mathcal{P}})$ contains the complete set of all logit vectors and their limiting points. The former corresponds to *non-boundary types*, whose choice model can be characterized by a standard MNL choice model, and the latter corresponds to *boundary types*, whose standard MNL parameters become unbounded, resulting infinite utility for some options and zero for the rest. Such broad search space $\text{Conv}(\overline{\mathcal{P}})$ in FW generates mixture compositions with a considerable number of the boundary types. More specifically, this problem is rooted to the linear subproblem solved at each *support finding step*, whose optimal solution always lies on the extreme point of the feasible region which contains many of these limiting choice probability vectors $\in \overline{\mathcal{P}} \setminus \mathcal{P}$ as its extreme points. Often, each limiting choice probability vector corresponds to an unbounded $\boldsymbol{\beta}_k$ and cause the learning outcome unusable for downstream applications. Though significant effort is made to justify the legitimacy of these boundary types in the paper, the convergence result only applies to the population's aggregate choice probability, i.e. $\hat{\mathbf{g}} \to \sum_k \alpha_k \boldsymbol{q}_k$, which is an Frank-Wolfe property. In other words, their method fails to recover each of the individual logit models ($\boldsymbol{\beta}_k$) and the corresponding mixture weights ($\alpha_k$) accurately.

To remedy this issue and seek convergence guarantees for the general MMNL estimation problem, we designed the Stochastic Subregion Frank-Wolfe (SSRFW) algorithm, as shown in Algorithm 1.

Note in the above algorithm, after $k$ iterations, we have generated $k$ choice probability vectors $\boldsymbol{q}^{(s)}$, $s = 1, \ldots, k$, all of which will be the output of the learning algorithm. The mixture weight

---

**Algorithm 1:** Stochastic Subregion Frank-Wolfe

---

**Input:** data $\mathbf{y}$, $\mathcal{Q}$ from Algorithm 2
**Initialization:** $k = 0$; $\boldsymbol{\alpha}^{(0)} = [1]$, a random $\mathbf{g}^{(0)} = \boldsymbol{q}^{(0)}$ chosen from $\mathcal{Q}$

1 **while** *stopping condition not met* **do**

2     $k \leftarrow k + 1$

3     Compute $\boldsymbol{q}^{(k)} = \underset{\boldsymbol{v} \in \mathrm{Conv}(\mathcal{Q})}{\arg\min} \left\langle \nabla \mathcal{L} \left( \mathbf{g}^{(k-1)} \right), \boldsymbol{v} - \mathbf{g}^{(k-1)} \right\rangle$

4     Compute $\boldsymbol{\alpha}^{(k)} = \underset{\boldsymbol{\alpha} \in \Delta_k}{\arg\min} \, \mathcal{L} \left( \alpha_0^{(k)} \mathbf{g}^{(0)} + \sum_{s=1}^{k} \alpha_s^{(k)} \boldsymbol{q}^{(s)} \right)$ [1]

5     Update $\mathbf{g}^{(k)} := \alpha_0^{(k)} \mathbf{g}^{(0)} + \sum_{s=1}^{k} \alpha_s^{(k)} \boldsymbol{q}^{(s)}$

6 **end**

**Output :** choice prob. $\boldsymbol{q}^{(0)}, \ldots, \boldsymbol{q}^{(k)}$
           mixture weights. $\boldsymbol{\alpha}^{(k)} \in \Delta_k \subset \mathbb{R}^{k+1}$

---

estimator is slightly different. $\boldsymbol{\alpha}^{(k)} \in \mathbb{R}^{k+1}$ represents the mixture weight vector for the $k+1$ choice probability vectors we have generated so far, where $\alpha_s^{(k)}$ refers to the $s$-th element in the vector, corresponding to the mixture associated with $\boldsymbol{q}^{(s)}$. In other words, the vector length of $\boldsymbol{\alpha}$ increases by 1 for each iteration and only the final vector $\boldsymbol{\alpha}^{(k)}$ is outputted from the algorithm.

The key different in SSRFW is that it takes in an additional input $\mathcal{Q}$ from the $\mathcal{Q}$ *Construction Algorithm* (discussed in detail in Section 3), which is the key to eliminate the possibility of producing *boundary types*. In particular, the new feasible region $\mathrm{Conv}(\mathcal{Q})$ is a subset of $\mathrm{Conv}(\overline{\mathcal{P}})$, where each element $\boldsymbol{q} \in \mathcal{Q}$ will be learned from data and is guaranteed to be within an $\epsilon$-ball of the true choice probability vector for some mixture type with high probability. Subsequently, this also ensures the SSRFW algorithm to recover the mixture weight for each latent class. Note that the extreme points of $\mathrm{Conv}(\mathcal{Q})$ by definition is a subset of $\mathcal{Q}$. Since we only care about the extreme points of the feasible region, we can safely ignore any points $\in \mathrm{Conv}(\mathcal{Q}) \setminus \mathcal{Q}$ and replace $\mathrm{Conv}(\mathcal{Q})$ with $\mathcal{Q}$ in Algorithm 1.

Since Frank-Wolfe will converge to the optimal solution [Jaggi, 2013], and as we will see in Section 4 that the optimal solution to problem (1) is an interior point of the feasible region $\mathrm{Conv}(\mathcal{Q})$, the stopping condition can simply be set as $\left\| \mathbf{g}^{(k)} - \mathbf{y} \right\| \leq \epsilon$.

## 3   $\mathcal{Q}$ Construction Algorithm

Our primary goal in this section is to learn the input set $\mathcal{Q}$ for SSRFW (Algorithm 1). In particular for this set $\mathcal{Q} = \{\hat{\boldsymbol{q}}_\ell\}_{l=1,\ldots,L}$, we require that $\forall \, \hat{\boldsymbol{q}}_\ell$, there exists some mapping $\pi : [L] \to [K]$ such that $\left\| \hat{\boldsymbol{q}}_\ell - \boldsymbol{q}_{\pi(\ell)} \right\| \leq \epsilon$, where $\boldsymbol{q}_{\pi(\ell)}$ is one of the ground truth logit vectors. Once this property holds, each of the SSRFW 's $\boldsymbol{q}$ outputs will also be $\epsilon$-close to some true choice probability vector as the algorithm always selects an extreme point of $\mathrm{Conv}(\mathcal{Q})$ in each iteration.

The two major components for the $\mathcal{Q}$ Construction Algorithm is first computing a distance score matrix (Section 3.1) and then creating subsamples that contain decision makers from only one mixture type through random seeding and non-uniform sampling (Section 3.2). Section 3.3 presents the algorithm and discusses the properties of the learning outcomes.

### 3.1   Distance Score Matrix

We first define a distance score matrix $S$, whose element $s_{ij}$ measures the dissimilarity between any two decision makers $i$ and $j$ in term of their choice decisions. Recall that each $x_i^{(t)}$ are realizations of i.i.d categorical random variables with PMF $\boldsymbol{q}_k$ if $i$ is from mixture $k$. Denote its associated empirical cumulative distribution function (CDF) as $F_T(x; i)$ when a total of $T$ decisions are observed. In

---

[1] $\Delta_k = \{(\alpha_1, \ldots, \alpha_{k+1}) \in \mathbb{R}^{k+1} \big| \sum_{i=1}^{k+1} \alpha_i = 1, \alpha_i \geq 0\}$

particular,

$$F_T(x; i) = \frac{1}{T} \sum_{t=1}^{T} \mathbb{1}_{\{x_i^{(t)} \leq x\}}, x \in [M]$$

Define the distance score function for each pair of decision makers as

$$s_{ij} := s(i, j) = ||F_T(x; i) - F_T(x; j)||_\infty$$

where $||\cdot||_\infty$ represents the infinity norm.

## 3.2 Subsample Construction

We now want to create a number of subsamples such that each subsample of decision makers contains only one mixture type with high probability. Each of these subsamples is generated following a procedure that contains a *random seeding* step, and a *subsampling* step. These steps are repeated $L$ times to create a $\mathcal{Q}$ set of size $L$.

**Random Seeding.** This step randomly samples a decision maker $i \in [N]$ from the population and can be done with simple uniform sampling technique. The selected decision maker is called a *seed*. We will generate $L$ seeds and later investigate how large $L$ needs to be in order to cover all mixture types in $\mathcal{Q}$.

**Subsampling Strategy.** This step generates an index set $I(i)$ for a seed $i$ such that, with high probability, decision makers whose indices belonging to this set are of the same type as $i$. $I(i)$ is called the *subsample originated from seed* $i$.

Given a selected seed $i$, we first calculate an *accepting probability* $p_{j|i}$ which determines the likelihood of another decision maker $j$ being accepted to the corresponding subsample $I(i)$. It is designed to be a monotonically decreasing function with respect to the distance score we defined in Section 3.1:

$$p_{j|i} = f\left(s(i, j)\right)$$

A simple example can be $p_{j|i} = 1 - s(i, j)$. Intuitively, when $s(i, j)$ is small, seed $i$ and decision maker $j$'s empirical CDF is close to each other, indicating there is a higher chance that they are of the same type and sharing the same choice probability $\boldsymbol{q}$. Subsequently, $p_{j|i}$ will be larger compared to a large value of $s(i, j')$ for another person $j'$, resulting we accept $j$ with higher probability than $j'$. This is consistent with our objective that we want each subsample to be composed of decision makers of the same type as the seed.

For implementation, we first initiate an empty index set $I(i)$ for the selected seed $i$. Then we repeat the following two steps until we reach a desired subsample size $n$: 1) draw a random sample from set $[N] \setminus I(i)$, 2) accept this sample into the index set with probability $p_{j|i}$ and reject with $1 - p_{j|i}$.

## 3.3 Algorithm and Properties

The remaining step is to obtain the set $\mathcal{Q}$ whose elements $\hat{\boldsymbol{q}}_\ell$ are computed as the average of historical choice decisions from decision makers in the subsamples $I_\ell$, i.e. $\hat{\boldsymbol{q}}_\ell = \frac{1}{nT} \sum_{i \in I_\ell} \sum_{t=1}^{T} Y_i^{(t)}$. Finally, we present the $\mathcal{Q}$ Construction Algorithm in Algorithm 2.

**Discussion** We next discuss a few properties of the two algorithms.

SSRFW returns the logit vectors $\boldsymbol{q}$ instead of the parameters $\boldsymbol{\beta}$. We can simply perform maximum likelihood estimation under the single MNL setting where, if the extreme point corresponding to subsample $\ell$ is selected during an SSRFW iteration, the log-likelihood is

$$l_\ell(\boldsymbol{\beta}) = -\sum_{t=1}^{T} \sum_{i \in I_\ell} \sum_{j \in [M]} Y_{ij}^{(t)} \left( \log \frac{\exp(\boldsymbol{\beta}^\top \boldsymbol{z}_j)}{\sum_{m \in [M]} \exp(\boldsymbol{\beta}^\top \boldsymbol{z}_m)} \right)$$

and we have $\hat{\boldsymbol{\beta}}_\ell = \arg\max_{\boldsymbol{\beta}} l_\ell(\boldsymbol{\beta})$. With the assumption of linear utility in $\boldsymbol{\beta}$, we can obtain the optimal solution using MLE (c.f. McFadden [1974]).

**Algorithm 2:** The $\mathcal{Q}$ construction algorithm

---

**Input :** score matrix $S$, number of subsamples $L$, subsample size $n$
**Initialization:** $\mathcal{Q} = \texttt{set}()$

**1 for** $\ell \leftarrow 1$ **to** $L$ **do**
**2**      Choose seed: $i \sim U(0, N)$
**3**      Initiate: $I_\ell = \texttt{set}()$
**4**      **while** $|I_\ell| \neq n$ **do**
**5**          $j \leftarrow \texttt{random\_sample}([N] \setminus I_\ell)$
**6**          Generate $u \sim U(0, 1)$
**7**          **if** $u < p_{j|i}$ **then**
**8**             $I_\ell.\texttt{add}(j)$
**9**          **end**
**10**      **end**
**11**      Compute $\hat{\boldsymbol{q}}_\ell = \dfrac{1}{nT} \sum_{i \in I_\ell} \sum_{t=1}^{T} Y_i^{(t)}$
**12**      $\mathcal{Q}.\texttt{add}(\hat{\boldsymbol{q}}_\ell)$
**13 end**
**Output:** $\mathcal{Q}$

---

In general, we prefer learning $\boldsymbol{\beta}_k$'s compared to $\boldsymbol{q}_k$'s. This makes the algorithm more robust when item attributes change over time which results in change in the $\boldsymbol{q}_k$'s whereas $\boldsymbol{\beta}_k$ values persist. To achieve this, we just need to add two additional steps after the index set is created:

- **MNL paramter estimation:** apply the MLE step outlined above; create $\mathcal{B} = \{\hat{\boldsymbol{\beta}}_\ell\}_{\ell=1,\ldots,L}$
- **Map $\mathcal{B}$ to $\mathcal{Q}$:** fix item attributes of current interest, calculate $\mathcal{Q} = \{\hat{\boldsymbol{q}} | \hat{\boldsymbol{q}} = \boldsymbol{q}(\hat{\boldsymbol{\beta}}_\ell), \hat{\boldsymbol{\beta}}_\ell \in \mathcal{B}\}$

Next, stochasticity in our algorithm comes from two parts, namely the realization of each decision makers' choices, as well as the randomness in the process of generating subsamples. Yet we will show that under appropriate assumptions, `SSRFW` can still recover all individual mixture parameters with high probability.

In addition, we do not impose any restriction on the subsamples created during $\mathcal{Q}$ construction to be mutually exclusive. As long as they only contain one single mixture type (i.e. with high purity), it creates a legitimate estimation of $\hat{\boldsymbol{q}}$ or $\hat{\boldsymbol{\beta}}$ with MLE.

Last but not least, this is a more robust strategy than clustering algorithms to segment the population. Not only do we not require the number of mixtures $K$ as a hyper-parameter, each decision maker's data being used more than once for the estimation can be thought of as a special bootstrap mechanism that has custom weights tailored to our objective in creating homogeneous subsamples.

## 4   Theory of the `SSRFW` Algorithm

Theorem 1 is the main result of this paper. To prove it, we will break down this section into two parts, where the first part will examine the provable convergence properties of the estimators, and the second part will focus on the sample complexity. We will outline the proof sketch but all detailed proofs can be found in the Appendix.

**Provable Convergence.**   Assume $K'$ is the total number of iterations that the algorithm has performed before reaching the stopping criteria. To simplify notation, we denote the generated outcome from $k$-th iteration $\boldsymbol{q}^{(k)}$ in `SSRFW` as $\hat{\boldsymbol{q}}_k$ to indicate they are the estimators and the final updated mixture weight corresponding to $\boldsymbol{q}^{(k)}$, $\alpha_k^{(K')}$, as $\hat{\alpha}_k$. In other words, we have $\mathbf{g}^{\texttt{SSRFW}} = \sum_{k'=1}^{K'} \hat{\alpha}_{k'} \hat{\boldsymbol{q}}_{k'}$, We use a similar notation $\mathbf{g}^{\texttt{FW}}$ for the original FW approach.

Figure 1 describes the necessary components and steps of proving the result. In particular, Property 1 and 2 (purple boxes) are adapted from existing results of the Frank-Wolfe framework [Jaggi, 2013]. In Appendix A.1.1, we prove Property 1 in Lemma 2. Property 2 follows from the fact that each of

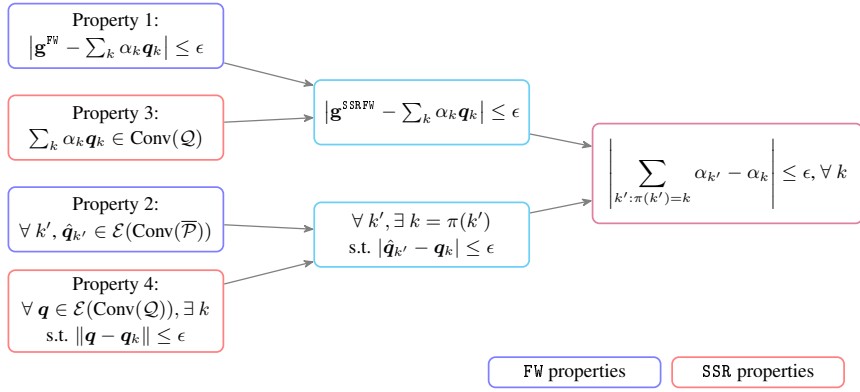

Figure 1: Proof sketch of Thoerem 1

the $\hat{q}_{k'}$, obtained by solving a linear optimization problem, has to be an extreme point (denoted by $\mathcal{E}(\cdot)$) of the feasible region.

On the other hand, Property 3 and 4 (red boxes) hold as a consequence of the stochastic subregion we constructed in Algorithm 2. Property 3 claims that with high probability, the ground truth aggregated choice probability vector is still an interior point of the shrunken feasible region $\text{Conv}(\mathcal{Q})$ (proof in Appendix A.1.2). Together with Property 1, this tells us that $|\mathbf{g}^{\text{SSRFW}} - \sum_k \alpha_k \boldsymbol{q}_k| \leq \epsilon$. That is to say the aggregated choice probability vector of SSRFW also converges to the ground truth. To achieve Property 4 from the $\mathcal{Q}$ construction algorithm, we need to supply enough of data, the details of which will be discussed in the sample complexity part. Combining Property 2 and 4, we have $\forall k', \exists k = \pi(k')$ s.t. $|\hat{\boldsymbol{q}}_{k'} - \boldsymbol{q}_k| \leq \epsilon$. Finally, by doing some algebraic manipulation, we can obtain $\left|\sum_{k':\pi(k')=k} \alpha_{k'} - \alpha_k\right| \leq \epsilon$.

**Sample Complexity.** Recall we use $T$ as the number of choice data records for each decision maker. Denote $T_{\min}$ as the minimum number required and $T_{\min} \geq 1$.

Define *in-types* to be the decision makers who share the same type as the seed, and *out-types* to be the ones of different types. We focus on the following three properties:

- probability of accepting the in-types,
- probability of rejecting the out-types,
- probability of covering all mixture types.

Consider a seed $i$, with type $k_i = k$. Assume there is a total of $m_k$ decision makers in the population that is of type $k$.

**Accepting In-Types.** Denote the empirical choice probability for the seed $i$ as $F_n$ and that for another decision maker $j$ as $G_m$, with $n$ and $m$ representing the number of independent price experiments from $i$ and $j$, respectively. Recall that $s(i,j) = ||F_n(x) - G_m(x)||_\infty = \sup_x |F_n(x) - G_m(x)|$. For exposition simplicity, we use $F$ and $G$ to denote $F(x)$ and $G(x)$ in the subsequent presentation if no confusion incurred.

**Theorem 2** (Sample Complexity I). *Assume $i$ and $j$ are of the same type. Define $p_{j|i} = 1 - s(i,j)$. $\forall \delta > 0, \epsilon > 0$, we can achieve $p_{j|i} > 1 - \epsilon$ with probability at least $1 - \delta$ with sample size $T_{\min} = \mathcal{O}(\frac{1}{\epsilon^2} \log\left(\frac{1}{\delta}\right))$.*

Theorem 2 indicates that with sufficient amount of data, there is a high probability to accept a in-type—who shares the same type as the seed—into the subsample if it is chosen from the population after the random draw.

**Rejecting Out-Types.** Assume a decision maker $j$ is of a different type from the seed $i$. We use $F$ and $G$ to denote their CDFs. Let $F_n$ and $G_m$ represent the corresponding empirical CDFs based on

samples drawn from the two different distributions with sample size $n$ and $m$, respectively. $s(i, j)$ is defined the same as before.

**Theorem 3** (Sample Complexity II). *Assume $F$ and $G$ correspond to the choice CDFs of two different types and $\sup_x |F(x) - G(x)| \geq \xi$. Let $F_n$ and $G_m$ be independent empirical distribution based on $m$ and $n$ i.i.d. samples drawn from $F$ and $G$, respectively. Denote $T_{min} = min\{m, n\}$. Define $p_{j|i} = 1 - s(i, j)$. $\forall \delta > 0, \epsilon > 0$, we can achieve $p_{j|i} < 1 - \xi + \epsilon$ with probability at least $1 - \delta$ with $T_{min} = \mathcal{O}(\frac{1}{\epsilon^2} \log \frac{1}{\delta})$.*

Theorem 3 complements Theorem 2 to rule out the out-types from being selected with a high probability. The more separable (larger $\xi$) their underlying distributions are, the lower accepting probabilities are.

**Lemma 1.** *For $\epsilon > 0$ and $0 < \delta < 1$, with minimum of choice records $T$ required by Theorem 2 and Theorem 3, $\forall \; \boldsymbol{q}_\ell \in \mathcal{Q}, \; \left\| \boldsymbol{q}_\ell - \boldsymbol{q}_{\pi(\ell)} \right\| \leq \epsilon$ with probability at least $1 - \delta$, where $\hat{\boldsymbol{q}}_\ell = \frac{1}{nT} \sum_{i \in I_\ell} \sum_{t=1}^T Y_i^{(t)}$.*

Lemma 1 states that with enough data samples, the $\mathcal{Q}$-construction algorithm achieves Property 4, as shown in Figure 1, which SSRFW's desired provable convergence builds upon. With Lemma 1, we conclude the proof for Theorem 1. The complete proof can be found in Appendix A.4.

As a final remark, we discuss the number of subsamples (i.e. $L$) needed in the $\mathcal{Q}$-construction process. We distinguish this from sample complexity as we are not requesting more data points with a larger $L$. Instead, we should view $L$ as a computational complexity factor. Recall that each moving direction derived from the *support finding step* is viewed as a learned logit vector of one mixture component. Therefore, we want the $\mathcal{Q}$-construction algorithm to include at least one $\hat{\boldsymbol{q}}_\ell$ for each mixture $k$ in order for it to be picked in SSRFW . This impose a requirement on $L$. Theorem 4 states that the expected number of the number of subsamples is controlled by the smallest mixture weight.

**Theorem 4.** *Assume $\alpha_1 \leq \alpha_2 \leq \cdots \alpha_K$. The expected number of subsamples $L$ we need to construct is bounded by $\frac{1}{\alpha_1} \log \frac{1}{\alpha_1}$.*

Intuitively, this means if we have one mixture component that is very under-represented, then we need to create more subsamples to ensure that one of the seeds belongs to this mixture.

# 5 Conclusion

In this work, we propose the SSRFW algorithm, which provides an end-to-end solution for learning MMNL models using historical choice data. This novel approach utilizes a carefully designed sampling method to construct a meaningful search space. Not only does it resolve the drawback of the original Frank-Wolfe approach with boundary-type issues but also enables us to obtain provable guarantees for the model estimates and sample complexity. Another major advantages of our algorithm is that we do not assume prior knowledge of the number of mixtures $K$ in the model. It is also more robust than traditional unsupervised approaches such as clustering method and EM.

In marketing science and operations research, MMNL provides a fundamental tool for demand prediction, which many downstream applications such as pricing and inventory position rely on. For instance, consider a pricing model that tries to maximize the revenue by solving the following optimization problem: $\max_{\boldsymbol{p}} \sum_{j=1}^M p_j \sum_{k=1}^K \alpha_k q_j(\boldsymbol{\beta}_k; \boldsymbol{p})$ where $p_j \geq 0, \forall j \in [M]$. This requires an accurate estimation of each individual MNL components. Due to page limit, we include our simulation studies and application in real-world setting in the appendix. We will also briefly discuss the requirements on the datasets in order to apply our algorithm.

## Acknowledgments and Disclosure of Funding

The research of Z. Yan was partly supported by Nanyang Technological University startup grant and MOE Academic Research Fund Tier 1 [Grant RG17/21] and Tier 2 [Grant MOE2019-T2-1-045] and NOL Fellowship grant [NOL21RP04].

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
