# A Proofs

## A.1 Proofs for Provable Convergence

### A.1.1 Proof of Property 1

**Lemma 2.** *Denote $\mathbf{g}^*$ as the optimal solution to the Stochastic Subregion Frank-Wolfe algorithm and $\mathbf{g}^{(k)}$ denote the $k$-th iterate generated by Algorithm 1. Then*

$$\mathcal{L}(\mathbf{g}^{(k)}) - \mathcal{L}(\mathbf{g}^*) \leq \frac{4}{k+2}$$

*for all $k \geq K$.*

**Proof**: This Lemma follows directly from the existing results of the original Frank-Wolfe algorithm and its variants [Jaggi, 2013], which states that for an optimization problem $\min_{x \in \mathcal{D}} f(\mathbf{x})$ where $f$ is a convex and continuously differentiable function and that the domain $\mathcal{D}$ is a compact convex set of any vector space, then for each $k \geq 1$, the iterates $\mathbf{x}^{(k)}$ of the fully-corrective Frank-Wolfe algorithm satisfy:

$$f(\boldsymbol{x}^{(k)}) - f(\boldsymbol{x}^*) \leq \frac{2 \cdot C_f}{k+2} \tag{2}$$

where $C_f$, defined as

$$C_f := \sup_{\substack{\boldsymbol{x},\boldsymbol{s} \in \mathcal{D} \\ \gamma \in [0,1] \\ \boldsymbol{r} = \boldsymbol{x} + \gamma(\boldsymbol{s} - \boldsymbol{x})}} \frac{2}{\gamma^2} \left( f(\boldsymbol{r}) - f(\boldsymbol{x}) - \langle \nabla f(\boldsymbol{x}), \boldsymbol{r} - \boldsymbol{x} \rangle \right),$$

is the *curvature constant*, which measures the "non-linearity" of function $f$ over domain $\mathcal{D}$. The type of the Frank-Wolfe we use in Algorithm 1 is precisely the fully-corrective variant in that we optimize for $\alpha$'s in each iteration.

**Claim 1.** $\mathcal{L}(\mathbf{g}; \mathbf{y}) = \|\mathbf{g} - \mathbf{y}\|^2$ *is a twice differentiable convex function. Conv$(\mathcal{Q})$ is a compact convex set.*

The first statement in Claim 1 is true by definition. The second statement can be shown by observing that $\mathcal{Q}$ is a finite set, hence compact, followed by the fact that convex hulls of compact set are compact.

For squared loss function $\mathcal{L}$ used in our model, Jagabathula et al. [2020] proved that $C_{\mathcal{L}} \leq 2$. The result of Lemma 2 follows by plugging $C_{\mathcal{L}}$ into Equation 2.

$\square$

Note that with enough sample, we also have $\left\| \mathbf{y} - \sum_{k=1}^{K} \alpha_k \boldsymbol{q}_k \right\| \leq \epsilon$ by the law of large numbers, which leads to $\mathbf{g}^{\texttt{SSRFW}}$ converges to $\sum_{k=1}^{K} \alpha_k \boldsymbol{q}_k$ with high probability. This shows that Frank-Wolfe can reach any tolerance level $\epsilon$ with enough number of iterations by setting appropriate stopping criteria.

### A.1.2 Proof of Property 3

**Lemma 3** (Wendel [1962]). *If $X_1, \ldots, X_n$ are i.i.d. random points in $\mathbb{R}^d$ whose distribution is symmetric with respect to the center $O$ and assigns measure zero to every hyperplane through $O$, then*

$$P_n^{(d)}(O \in Conv\{X_1, \ldots, X_n\}) = 1 - \frac{1}{2^{n-1}} \sum_{k=0}^{d-1} \binom{n-1}{k}$$

Lemma 3 is an interesting result from stochastic geometry, which states that if we randomly sample $n$ points in a $d$-dimensional ball, the probability that the convex hull formed using these points contains the center point can be computed using the above formula.

**Corollary 1.**
$\lim_{n \to \infty} P_n^{(d)}(O \in Conv\{X_1, \ldots, X_n\}) = 1$

**Proof**: When $n \geq 2d - 1$, $\binom{n-1}{k}$ is a monotonically increasing function of $k$ for $k = 0, \ldots, d - 1$. We then have

$$\sum_{k=0}^{d-1} \binom{n-1}{k} \leq d\binom{n-1}{d-1} \leq d\frac{(n-1)^d}{(d-1)!}$$

Therefore, when $n \geq 2d - 1$,

$$P_n^{(d)}(O \in \mathrm{Conv}\{X_1, \ldots, X_n\}) = 1 - \frac{1}{2^{n-1}}\sum_{k=0}^{d-1}\binom{n-1}{k} \geq 1 - \frac{d}{(d-1)!}\frac{(n-1)^d}{2^{n-1}}$$

Since $\lim_{n \to \infty} \frac{(n-1)^d}{2^{n-1}} = 0$, we get

$$\lim_{n \to \infty} P_n^{(d)}(O \in \mathrm{Conv}\{X_1, \ldots, X_n\}) = 1$$

$\square$

**Corollary 2.** *With high probability, $\mathrm{Conv}(\{\boldsymbol{q}_k\}_{1,\ldots,K}) \subseteq \mathrm{Conv}(\mathcal{Q})$*

The proof of Corollary 2 will be included in the proof of Theorem 1 in Appendix A.4. We will also quantify what "high probability" it is referring to. Corollary 2 establishes the fact that $\mathbf{g} = \sum_{k=1}^{K} \alpha_k \boldsymbol{q}_k \in \mathrm{Conv}(\{\boldsymbol{q}_k\}_{k=1,\ldots,K}) \subseteq \mathrm{Conv}(\mathcal{Q})$ with high probability. We illustrate this idea in Figure 2 for an intuitive understanding.

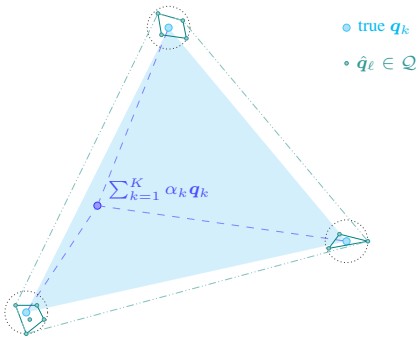

Figure 2: Constructed convex hull using logit vectors generated from subsamples

For each mixture type $k$, with enough data points, we have with probability $1 - \delta$, $\forall \ell$ such that $\pi(\ell) = k$, $\hat{\boldsymbol{q}}_\ell$ is within an $\epsilon$-ball of $\boldsymbol{q}_k$ given sufficient number of samples. In addition, according to Lemma 3, with high probability, such convex hull (small regions with green firm lines for each $k$) contains the ground truth choice probability vector $\boldsymbol{q}_k$, i.e. $\boldsymbol{q}_k \in \mathrm{Conv}(\{\boldsymbol{q}_\ell \in \mathcal{Q}|\pi(\ell) = k\})$. Since the true aggregated choice probability $\sum_{k=1}^{K} \alpha_k \boldsymbol{q}_k$ is a convex combination of $\boldsymbol{q}_k$'s (so it is in the blue shaded region) and $\bigcup_k \mathrm{Conv}(\{\boldsymbol{q}_\ell \in \mathcal{Q}|\pi(\ell) = k\}) \subset \mathrm{Conv}(\mathcal{Q})$, we have $\mathrm{Conv}(\mathcal{Q})$ encloses the blue region and $\sum_{k=1}^{K} \alpha_k \boldsymbol{q}_k$ is an interior point of $\mathrm{Conv}(\mathcal{Q})$.

### A.2 Proofs for Sample Complexity

We first extend the Dvoretzky-Kiefer-Wolfowitz (DKW) inequality with the following Lemma, where both CDFs in the inequality are empirical.

**Lemma 4.** *Let $F_n$ and $G_m$ be independent empirical distribution based on $m$ and $n$ i.i.d. samples drown from a common cumulative distribution $F(\cdot)$. Denote $min\{m, n\}$ as $T_{min}$. We have*

$$\mathbb{P}\left(\sup_x |F_n(x) - G_m(x)| > \epsilon\right) \leq 4\exp\left(-\frac{1}{2}T_{min}\epsilon^2\right)$$

**Proof**: Lemma 4 differs from the Dvoretzky-Kiefer-Wolfowitz (DKW) inequality in that it investigates the tail probability of the maximum difference between *two empirical* distributions. By DKW,

we know that

$$\mathbb{P}\left(\sup_x |F_n(x) - F(x)| > \epsilon\right) \le 2\exp\left(-2n\epsilon^2\right)$$

$$\mathbb{P}\left(\sup_x |G_m(x) - F(x)| > \epsilon\right) \le 2\exp\left(-2m\epsilon^2\right)$$

if $F_n$ and $G_m$ are empirical distributions of samples drawn from their true distribution function $F$ and $G$ respectively. We can show

$$\mathbb{P}\left(\sup_x |F_n - F| + \sup_x |G_m - F| > \epsilon\right) \tag{3}$$

$$\le \quad 1 - \mathbb{P}\left(\sup_x |F_n - F| \le \frac{\epsilon}{2} \cap \sup_x |G_m - F| \le \frac{\epsilon}{2}\right)$$

$$\le \quad 1 - \left(1 - 2\exp\left(-\frac{1}{2}n\epsilon^2\right)\right)\left(1 - 2\exp\left(-\frac{1}{2}m\epsilon^2\right)\right)$$

$$\le \quad 4\exp\left(-\frac{1}{2}T_{\min}\epsilon^2\right)$$

where the first inequality makes use of the fact that $\sup_x |F_n - F| + \sup_x |G_m - F| > \epsilon$ implies that either $\sup_x |F_n - F| > \frac{\epsilon}{2}$ or $\sup_x |G_m - F| > \frac{\epsilon}{2}$. The second inequality comes from the independent assumption between $F_n$ and $G_m$. On the other hand, we also have

$$\sup |F_n - F| + \sup |F - G_m| \tag{4}$$
$$\ge \quad \sup |F_n - F| + |F - G_m|$$
$$\ge \quad \sup |F_n - F + F - G_m|$$
$$= \quad \sup |F_n - G_m|$$

Combining (3) and (4), we can obtain

$$\mathbb{P}(\sup_x |F_n(x) - G_m(x)| > \epsilon)$$
$$\le \quad \mathbb{P}(\sup_x |F_n(x) - F(x)| + \sup_x |G_m(x) - F(x)| > \epsilon)$$
$$\le \quad 4\exp\left(-\frac{1}{2}T_{\min}\epsilon^2\right).$$

$\square$

### A.2.1 Proof of Theorem 2

**Proof**: For simpler notation, denote $s(i, j) = s$. Since $i$ and $j$ are of the same consumer type $k$, $F_n$ and $G_m$ are empirical distributions based on $n$ and $m$ samples drawn from the same distribution $q_k$. According to Lemma 4,

$$\mathbb{P}(p_{j|i} > 1 - \epsilon) = \mathbb{P}(s < \epsilon) \ge 1 - 4\exp(-\frac{1}{2}T_{\min}\epsilon^2)$$

Let $\delta = 4\exp(-\frac{1}{2}T_{\min}\epsilon^2)$, we have $T_{\min} = \mathcal{O}(\frac{1}{\epsilon^2}\log\frac{1}{\delta})$. $\square$

### A.2.2 Proof of Theorem 3

**Proof**: By definition,

$$\xi \le \quad \sup_x |F - G|$$
$$\le \quad \sup_x |F - F_n + F_n - G_m + G_m - G| \tag{5}$$
$$\le \quad \sup_x |F - F_n| + \sup_x |F_n - G_m| + \sup_x |G_m - G|$$

Therefore,

$$\mathbb{P}(\sup_x |F_n - G_m| \le \epsilon) \tag{6}$$

$$\le \quad \mathbb{P}(\sup_x |F - F_n| + \sup_x |G_m - G| > \xi - \epsilon)$$

$$\le \quad \mathbb{P}(\sup_x |F - F_n| > \frac{\xi - \epsilon}{2} \cup \sup_x |G - G_m| > \frac{\xi - \epsilon}{2})$$

$$= \quad 1 - \mathbb{P}(\sup_x |F - F_n| \le \frac{\xi - \epsilon}{2})\mathbb{P}(\sup_x |G - G_m| \le \frac{\xi - \epsilon}{2})$$

$$\le \quad 4\exp\left(-2T_{min}\left(\frac{\xi - \epsilon}{2}\right)^2\right)$$

where the first inequality is based on (5) and the second inequality makes use of the fact that $\sup_x |F_n - F| + \sup_x |G_m - F| > \xi - \epsilon$ implies that either $\sup_x |F_n - F| > \frac{\xi - \epsilon}{2}$ or $\sup_x |G_m - F| > \frac{\xi - \epsilon}{2}$. The equality comes from the independent assumption between $F_n$ and $G_m$. The last inequality is from the DWK inequality.

We want to restrain the sampling probability such that $p_{j|i} = 1 - s$ is within $\epsilon$-radius of the smallest possible sampling probability, which is $1 - \xi$, i.e.

$$\mathbb{P}(1 - s < 1 - \xi + \epsilon) = \mathbb{P}(s > \xi - \epsilon)$$
$$= 1 - \mathbb{P}(s \le \xi - \epsilon)$$
$$\ge 1 - 4\exp\left(-2T_{min}\left(\frac{\epsilon}{2}\right)^2\right)$$

as a result of Equation 6.

Let $\delta = 4\exp(-\frac{1}{2}T_{min}\epsilon^2)$. Then we have $T_{min} = \mathcal{O}(\frac{1}{\epsilon^2}\log\frac{1}{\delta})$. $\qquad\square$

**Additional discussion.** We further discuss the impact of $M$ on the sample complexity. As discussed above, we mainly relate the number of time periods we need to collect the data for, i.e. $T$, and $\epsilon$ and $\delta$ is through the DKW inequality. To put things simple, consider the case where we want to bound the probability that the supremum of the discrepancy between the empirical CDF and the true CDF larger than $\epsilon$ to be no bigger than $\delta$, as follows:

$$\mathbb{P}(\sup_{x \in [M]} |F_T(x) - F(x)| > \epsilon) \le 2e^{-2T\epsilon^2} \overset{\text{set to}}{=} \delta$$

This gives $T = \frac{1}{2\epsilon^2}\log\frac{2}{\delta}$. Define a random variable $U = \sup_{x \in [M]} |F_T(x) - F(x)|$. For a larger $M'$, similarly define $V = \sup_{x \in [M']} |F_T(x) - F(x)|$. Since the set that the supremum is taken is larger for $V$, we have $V \ge U$. Subsequently, we have $\mathbb{P}(V > \epsilon) \ge \mathbb{P}(U > \epsilon)$.

Assume $\mathbb{P}(V > \epsilon) = \delta + \xi$ for some $\xi \ge 0$. The relationship between $T$ and $\epsilon$ and $\delta$ is thus $T = \frac{1}{2\epsilon^2}\log\frac{2}{\delta + \xi}$ for the $M'$ case. Now if we want to bound $\mathbb{P}(V > \epsilon)$ by $\delta$, instead of $\delta + \xi$, we need to further increase the value of $T$, hence increase the sample complexity.

This analysis can be generalized directly to the two-sample DKW inequality for both in-type and out-type bounds and suggests an impact of $M$ on the sample complexity. This agrees with the intuition that for a larger choice set, we need to collect more data since otherwise the empirical choice distribution over the set will be very sparse, making the score function computed based on the empirical CDF less credible. However, we think such impact, compared to $\epsilon$ and $\delta$, is rather implicit as it is enclosed in the sup function.

## A.3 Proof of Theorem 4

**Proof**: As in Algorithm 1, we use $L$ as the number of subsamples we need to construct. Let $L_k$ be the number of subsamples needed to hit the $k$-th mixture type after $k - 1$ types of seeds have been selected. We have $L = L_1 + \cdots + L_K$.

We first construct a simple and fake scenario where we have $K'$ mixture types with each mixture weight equal to $\alpha_1$, i.e. $K' = \frac{1}{\alpha_1}$. Similarly, we can define $L'$ and $L'_k$ as above and also have

$L' = L'_1 + \cdots + L'_{K'}$. Think of $L'$ and $L'_k, k = 1, \ldots, K'$ as random variables and we know the probability of selecting a seed from a new type $k$ is $p_k = \frac{K'-k+1}{K'}$ since in the fake scenario, each type has the same probability $\alpha_1$ of being chosen. This tells us that $L'_k$ has a geometric distribution with expectation $\frac{1}{p_i} = \frac{K'}{K'-k+1}$.

By the linearity of expectations we have

$$
\begin{aligned}
\mathbb{E}[L'] &= \mathbb{E}[L'_1 + L'_2 + \cdots + L'_{K'}] \\
&= \mathbb{E}[L'_1] + \mathbb{E}[L'_2] + \cdots + \mathbb{E}[L'_{K'}] \\
&= \frac{K'}{K'} + \frac{K'}{K'-1} + \cdots + \frac{K'}{1} \\
&= K' \cdot \left( \frac{1}{1} + \frac{1}{2} + \cdots + \frac{1}{K'} \right) \\
&= K' \cdot H_{K'}
\end{aligned}
$$

where $H'_K$ is the $K'$-th harmonic number. Using the asymptotics of the harmonic numbers, we get

$$
\mathbb{E}[L'] \approx K' \log(K') = \frac{1}{\alpha_1} \log(\frac{1}{\alpha_1})
$$

Since $\alpha_1 \leq \cdots \leq \alpha_K$, we know $K' \geq K$. On the other hand, we know the $\mathbb{E}[L_k] \leq \mathbb{E}[L'_k]$ since there is a higher probability of choosing any mixture type $k \geq 2$, due to the same reason, i.e. $\alpha_k \geq \alpha_1$. Therefore, we have $\mathbb{E}[L] \leq \frac{1}{\alpha_1} \log(\frac{1}{\alpha_1})$

We can further characterize the probability of event $\mathcal{H}$, which describes the event that all mixture types are included in the constructed set $\mathcal{Q}$ by creating $L$ subsamples.

**Claim 2.** *For any $\delta > 0$, we have $\mathbb{P}(\mathcal{H}_L) \geq 1 - \delta$ with $L$ chosen according to the criteria described below.*

**Proof:** Denote $Z_k^L$ as the event that $k$-th mixture type is not being chosen as seed in the $L$ trials. Similarly, we can define $Z'^L_k$ for the fake scenario as described above. We then have

$$
\mathbb{P}(Z_k^L) = (1 - \alpha_k)^L \leq \left( 1 - \frac{1}{K'} \right)^L = \mathbb{P}(Z'^L_k) \leq e^{-\frac{L}{K'}}
$$

Denote $W_k$ as the event that the convex hull formed by the set $\{\hat{q}_\ell = \frac{1}{nT} \sum_{i \in I_\ell} \sum_{t=1}^T Y_i^{(t)} | \pi(i) = k\}$ for mixture type $k$ covers the true choice probability vector. Note that each $\frac{1}{T} \sum_{t=1}^T Y_i^{(t)}$ can be viewed as a sample mean of $q_k$ and by central limit theorem, it is symmetric with respect to $q_k$, hence so are the $\hat{q}_\ell$'s. According to Lemma 3 and Corollary 1,

$$
\begin{aligned}
\mathbb{P}(W_k) &= 1 - \frac{1}{2^{L_k-1}} \sum_{i=0}^{d-1} \binom{L_k - 1}{i} \\
&\geq 1 - \frac{d}{(d-1)!} \frac{(L-1)^d}{2^{L-1}} \qquad \text{when } L \geq 2d - 1
\end{aligned}
$$

Putting everything together, we have

$$
\mathbb{P}(\mathcal{H}) = \left( 1 - \mathbb{P}(\cup_{k=1}^K Z_k^L) \right) \mathbb{P}(\cap_{k=1}^K W_k) \tag{7}
$$

$$
\geq \left( 1 - \mathbb{P}(\cup_{k=1}^{K'} Z_k^L) \right) \left( 1 - \frac{1}{2^{L-1}} \sum_{i=0}^{d-1} \binom{L-1}{i} \right)^K \tag{8}
$$

$$
\geq \left( 1 - \frac{1}{\alpha_1} e^{-L\alpha_1} \right) \left( 1 - \frac{d}{(d-1)!} \frac{(L-1)^d}{2^{L-1}} \right)^K \tag{9}
$$

For any $\delta > 0$, we can then choose $L$ such that $1 - \delta \leq$ RHS of Equation (11) and $L \geq 2d - 1$.

$\square$

## A.4 Proof of Theorem 1

We first discuss Corollary 2 and what "high probability" refers to.

Denote $W_k$ as the event that the convex hull formed by the subsamples for a mixture type $k$ covers the true choice probability vector, as in Appendix A.3. If we have subsampled all mixture types and for each type $k$, event $W_k$ occurs, we can obtain $\boldsymbol{q}_k \in \text{Conv}(\mathcal{Q})$. Subsequently, we have $\text{Conv}(\{\boldsymbol{q}_k\}_{1,\dots,K}) \subseteq \text{Conv}(\mathcal{Q})$.

On the other hand, we have already analyzed the probability for event that *subsampled all mixture types and $Y_k$ occurs $\forall k$* to occur, which is precisely $\mathcal{H}_L$ as defined above. Specifically, it happens with probability $\geq \left(1 - \frac{1}{\alpha_1} e^{-L\alpha_1}\right) \left(1 - \frac{d}{(d-1)!} \frac{(L-1)^d}{2^{L-1}}\right)^K$. As $L$ increases, this number quickly increases to 1. This completes the proof of Corollary 2.

Finally, we combine all the results above and prove the provable convergence part in Theorem 1.

**Proof**: As illustrated in Figure 1, we want to show with probability $\geq 1 - \delta$ we have

> **S.1** $|\mathbf{g}^{\texttt{SSRFW}} - \sum_k \alpha_k \boldsymbol{q}_k| \leq \epsilon$
> **S.2** $\forall\, k',\exists\, k = \pi(k')$ s.t. $|\hat{\boldsymbol{q}}_{k'} - \boldsymbol{q}_k| \leq \epsilon$
> **S.3** $\left|\sum_{k':\pi(k')=k} \alpha_{k'} - \alpha_k\right| \leq \epsilon$

First, **S.1** is proved by Lemma 2 (Property 1: $|\mathbf{g}^{\texttt{FW}} - \sum_k \alpha_k \boldsymbol{q}_k| \leq \epsilon$) and Corollary 2 (Property 3: $\sum_k \alpha_k \boldsymbol{q}_k \in \text{Conv}(\mathcal{Q})$), together with the fact that $\mathbb{P}\left(\left|\frac{1}{T}\sum_{t=1}^{T}\mathbf{y}^t - \sum_k \alpha_k \boldsymbol{q}_k\right| \leq \epsilon\right) > 1 - \delta$ by central limit theorem. Since $\sqrt{T}(\frac{1}{T}\sum_{t=1}^{T}\mathbf{y}^t - \sum_k \alpha_k \boldsymbol{q}_k) \xrightarrow{d} \mathcal{N}(0, \sigma^2)$, number of samples required is also in the order of $\frac{1}{\epsilon^2}\log(\frac{1}{\delta})$.

Second, **S.2** is also a combined result by Frank-Wolfe's solving a linear program as an intermediate step (Property 2: $\forall\, k',\, \hat{\boldsymbol{q}}_{k'} \in \mathcal{E}(\text{Conv}(\overline{\mathcal{P}}))$, $\mathcal{E}(\cdot)$ denoting the extreme point set of the input region) and by construction using Algorithm 2 (Property 4: $\forall\, \boldsymbol{q} \in \mathcal{E}(\text{Conv}(\mathcal{Q})),\exists\, k$ s.t. $\|\boldsymbol{q} - \boldsymbol{q}_k\| \leq \epsilon$).

Subsequently, we can show **S.3**. Denote $K'$ as the number of mixtures output by the $\texttt{SSRFW}$ algorithm. Using **S.1**, we first write

$$\left|\sum_{k'=1}^{K'} \hat{\alpha}_{k'} \hat{\boldsymbol{q}}_{k'} - \sum_{k=1}^{K} \alpha_k \boldsymbol{q}_k\right| \leq \epsilon' \tag{10}$$

According to **S.2**, $\exists\, \pi$ such that $\pi(k') = k$ and we can write $\hat{\boldsymbol{q}}_{k'} = \boldsymbol{q}_{\pi(k')} + \boldsymbol{\epsilon}'$ where $|\boldsymbol{\epsilon}'| \leq \epsilon'$. Rearranging Eqn. (10) gives

$$\left|\sum_{k=1}^{K} \boldsymbol{q}_k \left(\sum_{k':\pi(k')=k} \hat{\alpha}_{k'} - \alpha_k\right) + \sum_{k'=1}^{K'} \hat{\alpha}_{k'} \boldsymbol{\epsilon}'\right| \leq \epsilon' \tag{11}$$

By triangle inequality, we get

$$\left|\sum_{k=1}^{K} \boldsymbol{q}_k \left(\sum_{k':\pi(k')=k} \hat{\alpha}_{k'} - \alpha_k\right)\right| - \epsilon' \leq \epsilon'$$

Since $\boldsymbol{q}_k$ is some arbitrary non-zero vector, we must have $\left(\sum_{k':\pi(k')=k} \hat{\alpha}_{k'} - \alpha_k\right) \leq 2\epsilon', \forall\, k$, which completes **S.3** by letting $\epsilon = 2\epsilon'$.

Note the above result holds assuming $K' \geq K$. To see why this is always the case, consider the linear system $Q\boldsymbol{x} = [\boldsymbol{q}_1 \boldsymbol{q}_2 \dots \boldsymbol{q}_K][x_1, x_2, \dots, x_K]^\top = \mathbf{g}$, where $Q \in \mathbb{R}^{M \times K}$ and $\mathbf{g} = \sum_{k=1}^{K} \alpha_k \boldsymbol{q}_k$. According to Definition 1, $D_{KS}(\boldsymbol{q}_k, \boldsymbol{q}_{k'}) \geq \epsilon$, we know that all $\boldsymbol{q}_k$'s are linearly independent. Since $M \geq K$, $\text{rank}(Q) = \text{rank}(Q|\mathbf{g}) = K$. The linear system has a unique solution that $\boldsymbol{x} = \boldsymbol{\alpha}$, where all $x_k$'s are non-zero. On the other hand, we have $\mathbf{g}^{\texttt{SSRFW}} = \sum_{k'=1}^{K'} \hat{\alpha_{k'}} \hat{\boldsymbol{q}}_{k'}$ and $\|\mathbf{g}^{\texttt{SSRFW}} - \mathbf{g}\| \leq \epsilon$. Assume $K' < K$, then upto a difference of $\epsilon$, the linear system $\hat{Q}\boldsymbol{x} = \mathbf{g}$, where $\hat{Q} = [\hat{\boldsymbol{q}}_1 \hat{\boldsymbol{q}}_2 \dots \hat{\boldsymbol{q}}_{K'}] \in$

$\mathbb{R}^{M \times K'}$, is inconsistent. In other words, we will not be able to obtain a $\mathbf{g}^{\mathtt{SSRFW}}$ that is $\epsilon$-close to $\mathbf{g}$, making it impossible to reach the stopping condition in the $\mathtt{SSRFW}$ algorithm. Therefore, the algorithm will keep going for more iterations, until we have at least $K' = K$.

Finally, according to Theorem 2, Theorem 3, the sample complexity is $\mathcal{O}(\frac{1}{\epsilon^2} \log(\frac{1}{\delta}))$.

$\square$

**Additional discussion.** In the proof, we showed that number of mixtures returned by $\mathtt{SSRFW}$, $K'$, is at least the ground truth number of mixtures, $K$. A natural question to ask is that how the misaligned number of mixtures affect the learning result, if $K' \neq K$. In many situations, this would not be a problem.

Consider the case that $\exists k_1, k_2$, such that $\pi(k_1) = \pi(k_2) = k$ while the rest are all one-to-one mapping. According to Theorem 1, we have $\mathbb{P}(|\boldsymbol{q}_{k_i} - \boldsymbol{q}_k| < \epsilon) \geq 1 - \delta$, for $i = 1, 2$ and $\mathbb{P}(|\alpha_{k_1} + \alpha_{k_2} - \alpha_k| < \epsilon) \geq 1 - \delta$. We can view the ground-truth MMNL model as an $(M+1)$-MNL model, where the original $k$-th mixture is now divided into two MNL components which share the identical logit parameters, where one of them has mixture weight $\alpha_{k_1}$ and the other one $\alpha_k - \alpha_{k_1}$. It is not hard to see that for the first component, we learned the correct mixture weight with an $\epsilon$-close logit vector $\hat{\boldsymbol{q}}_{k_1}$ while for the second, the mixture weight is off by at most $\epsilon$ with an $\epsilon$-close logit vector $\hat{\boldsymbol{q}}_{k_2}$.

Finally, we give some comment on the mapping function $\pi$. Note that we do not need this information other than using it as a tool in the proofs, though we can design heuristics to learn the mapping. In real world applications, we do not know the ground truth parameters, so we cannot derive such mapping anyways. On the other hand, if we run the algorithm multiple times, we will get different results due to the stochasticity embedded in the algorithm. In general, the lexicographic order of the mixtures is not important and can be reordered arbitrarily.

# B Numerical Experiments

In this section, we demonstrate the performance of our algorithm using numerical simulation and on the Nielsen Consumer Panel Data (https://www.chicagobooth.edu/research/kilts/datasets/nielsenIQ-nielsen).

## B.1 Simulation Study

Consider in a setting where there are $N$ consumers (i.e. decision makers) who make purchase decisions among a set of $M$ products for $T$ periods. Working with synthetic data allows us to measure the model performance regarding consumer type recovery since we have the knowledge of the underlying ground truth.

**Data Generation**  As an example experiment, we use following parameters: $N = 2000, K = 5, M = 10, d = 10$. We vary the total number of experiment epochs from $T = 5$ to 300. Mixture types are indexed by capital letters, i.e., "A", "B", "C", "D", "E", and products are indexed by numbers from 1 to 10. In the experiment, we also include an offset option, allowing the consumer to choose to not buy anything.

Features of different options and preference vector $\beta_k$'s are randomly generated in the interval $[-1, 1]$. $\alpha_k$'s are randomly generated such that $\sum_k \alpha_k = 1$ and the minimum mixture proportion $\min_k \alpha_k \geq \frac{1}{K+3}$ to ensure that not a particular type is under-represented. We set $L = 75$.

**Subsample Purity**  We first examine the purity of each subsample obtained. The reason we are interested in this quantity is that if a constructed subsample has high purity, it means that the empirical $\hat{q}$ we obtained from the subsample contains only one mixture type and thus can provide an estimate for the logit vector for that type with high accuracy. Figure 3 shows the average subsample purity with respect to the total number of experiment epochs. We can see that the constructed subsamples can achieve 90% purity with as few as 30 experiment epochs, and quickly reaches 99% around $T = 150$.

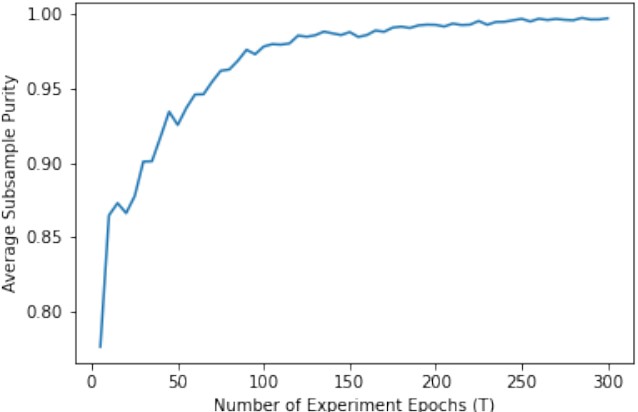

Figure 3: Subsample Purity Distribution

**Quality of $\mathcal{Q}$**  Next we evaluate the quality of the constructed set $\mathcal{Q}$ after applying Algorithm 2 to the simulated data. According to SSRFW, the estimated choice probability outcome is essentially a subset of these candidate vectors in $\mathcal{Q}$—the ones emitted by SSRFW at each iteration. Therefore, the higher the quality of $\mathcal{Q}$—in the sense it is concentrated near the ground truth—the better mixture estimation we can obtain using the learning algorithm SSRFW.

We claim each subsample is a representation of a particular mixture type in the ground truth if the majority of the selected consumers in that subsample are of that type. For instance, if a subsample contains 47 of type A, 2 of type B, and 1 of type E, we will evaluate the estimates $\hat{q}$ from this subsample against $\hat{q}_A$.

In Figure 4, we plot the distribution of choice probability values with respect to each product in the option set for all mixture types, with Figure 4a showing the result with 50 experiment epochs and Figure 4b with 300 experiment epochs.

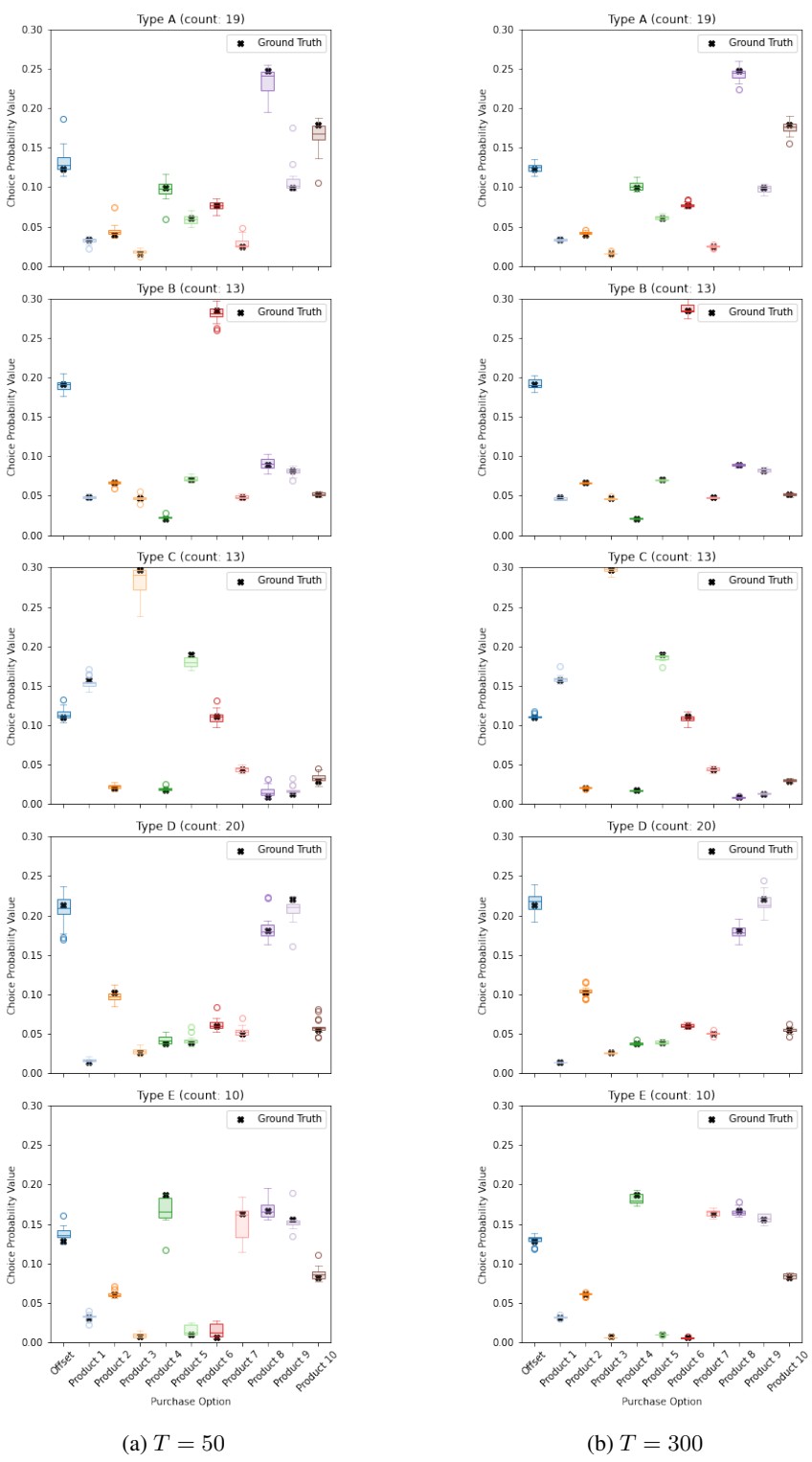

(a) $T = 50$  (b) $T = 300$

Figure 4: Choice Probability Estimation Result for All Consumer Types

These figures show that the estimated choice probability values are very concentrated near the ground truth when $T = 300$ and are also reasonably good even with a small $T = 50$, which we will proceed with for the following evaluation.

**Mixture Recovery**   After feeding the constructed $\mathcal{Q}$ to Algorithm 1, `SSRFW` generates 8 mixture types in terms of logit vectors (as expected, larger than the ground truth number of mixtures). We now want to compare these generated logit vectors against the ground truth. We refer to the mapping $\pi(j) = i$ as the "closest type" between a generated type $j$ and an ground truth (GT) type $i$ again using the majority rule mentioned above.

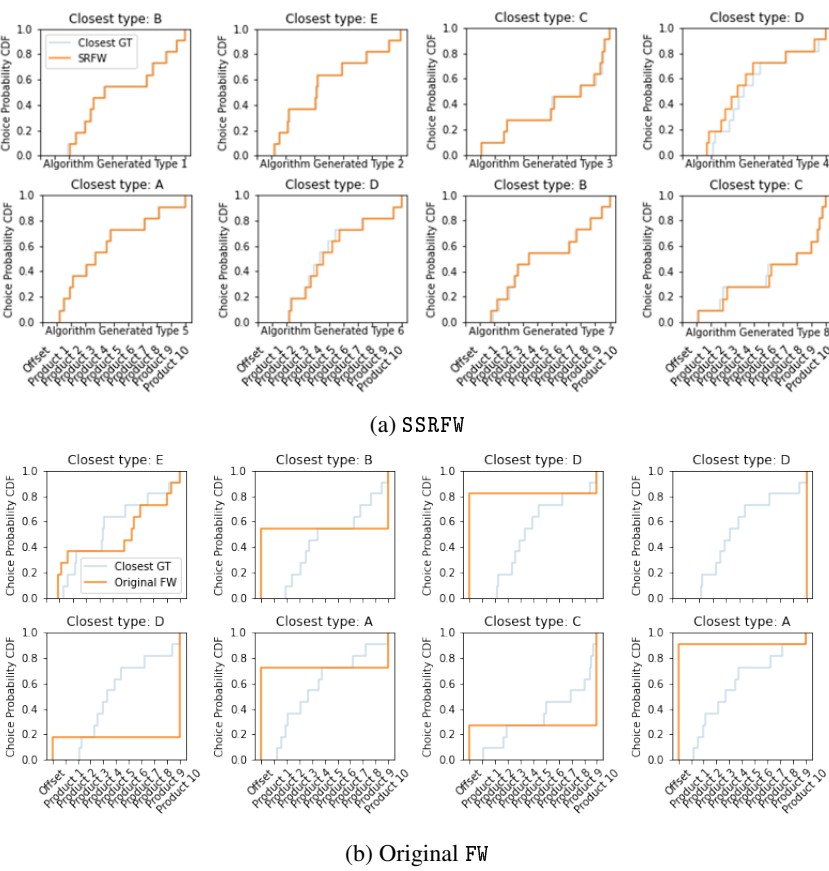

(a) `SSRFW`

(b) Original `FW`

Figure 5: Comparison of cumulative distribution of choice probability vectors

Figure 5 compares $j$ and $\pi(j)$ using the cumulative distribution (CDF) of the choice probability vectors, with orange lines representing learning outcome and blue lines the ground truth. We can see that the results from `SSRFW` are very close to the true mixture CDFs (Figure 5a). In contrast, the original `FW` algorithm [Jagabathula et al., 2020] does not recover the true choice probabilities (Figure 5b), which are mostly *boundary-types* that correspond to the extreme points of the feasible region to the individual LP subproblem in each FW iteration.

We further compare the mixture weight estimates $\alpha$'s in Table 1. From the table, we can see that if we have a one-to-one mapping (such as A and E), the $\alpha$ estimates from the algorithm are close to the true values. If we have many-to-one mappings, i.e., the algorithm outputs multiple mixtures for the same latent class, the sum of the estimated mixture weights is also close to the true values of each mixture.

**A Further Comparison with Original FW and EM**   We define another performance measure as the weighted average distance between the predicted choice probabilities and its closest ground truth, i.e. $\| \sum_k \alpha_k \boldsymbol{q}_k - \sum_k \hat{\alpha}_k \sum_{\pi(i)=k} \hat{\boldsymbol{q}}_i^{\mathrm{Alg}} \|$, where $\alpha_k$ and $\boldsymbol{q}_k$ are ground truth and Alg is one of `FW`, `SSRFW`, and `EM`. We vary the number of experiment epochs $T$ to examine how additional data can help improve the performance of MMNL learning algorithms.

Table 1: Estimation of Mixture Proportion $\alpha$

| Type | True $\alpha$ | $\hat{\alpha}^{\texttt{SSRFW}}$ | Type-wise Sum |
|------|------|------|------|
| A | 0.2000 | $\hat{\alpha}_5$: 0.1904 | 0.1904 |
| B | 0.2364 | $\hat{\alpha}_1$: 0.0713
$\hat{\alpha}_7$: 0.1607 | 0.2320 |
| C | 0.1636 | $\hat{\alpha}_3$: 0.0741
$\hat{\alpha}_8$: 0.0757 | 0.1498 |
| D | 0.2182 | $\hat{\alpha}_4$: 0.0387
$\hat{\alpha}_6$: 0.2143 | 0.2530 |
| E | 0.1818 | $\hat{\alpha}_2$: 0.1748 | 0.1748 |

In Figure 6a, we can see that the total discrepancy is consistently smaller using the $\texttt{SSRFW}$ compared to the original $\texttt{FW}$, which further justifies the good performance of our algorithm in predicting individual mixtures. In Figure 6b, we can see that both $\texttt{SSRFW}$ and EM benefit from having more repetitive choice data, especially when $T \leq 30$, which agrees with the observations we learned from Sample Purity and Quality of $\mathcal{Q}$ part.

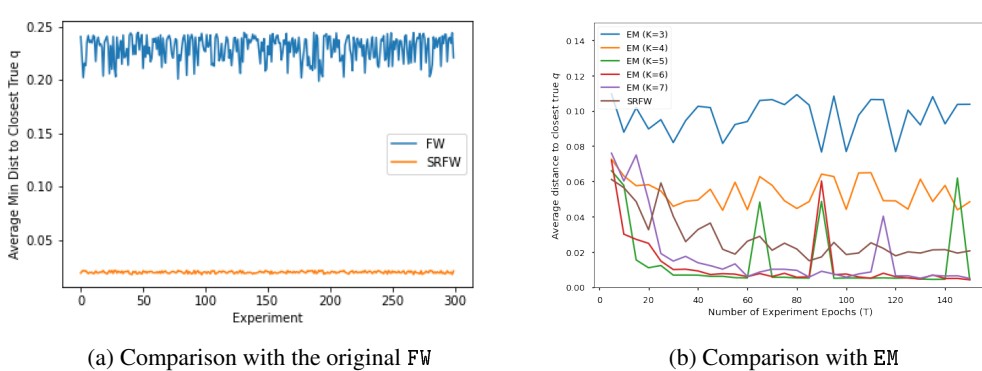

(a) Comparison with the original $\texttt{FW}$        (b) Comparison with $\texttt{EM}$

Figure 6: Performance comparison with other algorithms

Since EM requests the number of mixtures as an additional hyperparameter, we compare the performance of $\texttt{SSRFW}$ to EM with various different hyperparameter $\tilde{K}$ values, as shown in Figure 6b. We can see from the plot that when $\tilde{K} < K$, $\texttt{SSRFW}$ usually outperforms the EM algorithm. On the other hand, when $\tilde{K} >= K$, EM achieves a marginally better performance. However, in general we do not have this information, therefore we adopted the conventional strategy (c.f. Train [2008]) to use AIC/BIC to determine the best $\tilde{K}$, which corresponds to the lowest of these two criteria respectively. Both AIC/BIC measure the relative quality of statistical models for a given dataset, by balancing the trade-off between goodness of fit and the simplicity of the model and are commonly used in model selection. In our experiment, $K = 3$ gives the lowest value for both AIC and BIC. This indicates we should choose $K = 3$, yet it will result in worse estimation quality as shown in Figure 6b.

Last but not least, we can see that the major issue of EM is that it suffers from its instability. In reality, there is no way to tell whether the EM method is stuck at local optimum vs global optimum. In contrast, our algorithm is designed to give theoretical guarantees on the estimators. In summary, the combination of $\texttt{SSRFW}$'s theoretical guarantees on the learning result, sample complexity analysis, as well as the capability to recover individual level of mixture parameters makes the unique contribution of this work.

One limitation in our framework is related to data collection. For each decision maker, we require a series of historical choice data over a fixed set of options in order to get a reliable empirical cumulative distribution of their choice probability vector. However, this problem is not unique to our framework and EM also requires the same type of data. Thanks to the high digitalization level of many industries,

we can see that such data are widely available in a variety of settings, ranging from training NLP models with mixture of topics of text corpus to learning consumer preferences via data that record their purchases over some product sets, which we will discuss next.

## B.2 Experiments on Nielsen Consumer Panel Data

In this section, we demonstrate how we have applied `SSRFW` to the Nielsen Consumer Panel data. This comprehensive dataset is provided by the Kilts Center for Marketing at the University of Chicago Booth School of Business, NielsenIQ, and Nielsen. It contains panelists (i.e., households) purchase decisions on grocery items included in the NielsenIQ food and nonfood departments (roughly 1.4M UPC codes) dated back to 2004 with regular annual updates. The panel size varies from 40K to 60K and the characteristics include product description, brand, multipack, size, etc. This panel data is widely used for longitudinal studies in marketing science.

### B.2.1 Data Curation

We consider applying the algorithm to a substitute set of products under a particular category. This is a realistic setting as consumers usually choose one item from the substitution set. We curated data for six different categories, including yogurt, cereal, snack, candy, soft drinks and pet food and provide some summary statistics in Table 2.

Table 2: Nielsen case study: categories and data information

| Category | Panel size | Number of features | Average # purchases |
|----------|------------|--------------------|--------------------|
| yogurt | 1443 | 9 | 178 |
| pet-food | 2451 | 8 | 403 |
| candy | 1499 | 14 | 127 |
| cereal | 1085 | 13 | 96 |
| snack | 665 | 16 | 61 |
| soft-drinks | 412 | 12 | 209 |

### B.2.2 Experiment Setup

We cannot evaluate the model performance the same as in simulation studies since we no longer have the ground truth knowledge. Instead, we split the data into a training and test set, with the former used for learning the model parameters and the latter for evaluation. Specifically, we apply our algorithm to the training set, and use the learned parameters to compute the theoretical aggregated market share $\sum_k \hat{\alpha}_k \hat{\boldsymbol{q}}_k$. Then we compute its distance to the aggregated market share of the test set, i.e. $\| \sum_k \hat{\alpha}_k \hat{\boldsymbol{q}}_k - y^{\text{test}} \|$. The assumption is, if we have similar mixture composition in the training and test set, then the estimated parameter values from the training set should yield aggregated choice probability values close to that of the test set. To avoid randomness in the data split, we used a 10-fold cross validation, with the entire process repeated for five times.

### B.2.3 Results

Figure 7 plots the distribution of $\| \sum_k \hat{\alpha}_k \hat{\boldsymbol{q}}_k - y^{\text{test}} \|$ from the repeated runs of both algorithms. We can see that `SSRFW` in general outperforms the original FW algorithm in that the discrepancy is distributed close to 0.

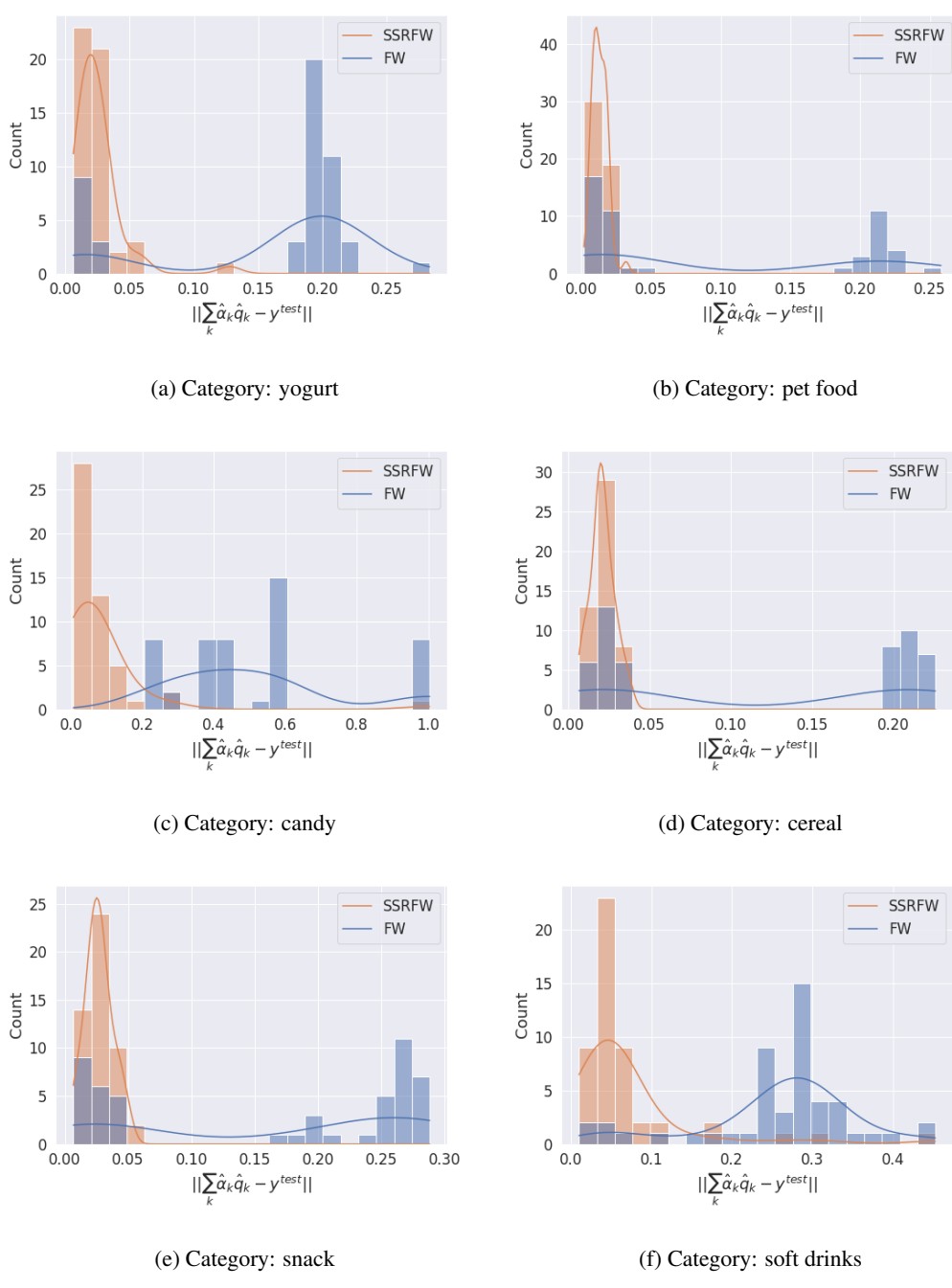

(a) Category: yogurt

(b) Category: pet food

(c) Category: candy

(d) Category: cereal

(e) Category: snack

(f) Category: soft drinks

Figure 7: $\|\sum_k \hat{\alpha}_k \hat{\boldsymbol{q}}_k - y^{\text{test}}\|$ for the six product categories

Figure 8 plots the deviation of product-level choice probability values from the test set, $\frac{|\sum_k \hat{\alpha}_k \hat{q}_{kj} - y_j^{\text{test}}|}{y_j^{\text{test}}}$, for $j = 1, 2, 3, 4, 5$. The light orange horizontal line indicates a zero deviation and we observe that the predicted aggregated choice probability per product level from SSRFW is more concentrated around zero than the original FW. In addition, there is a smaller variance with respect to different runs.

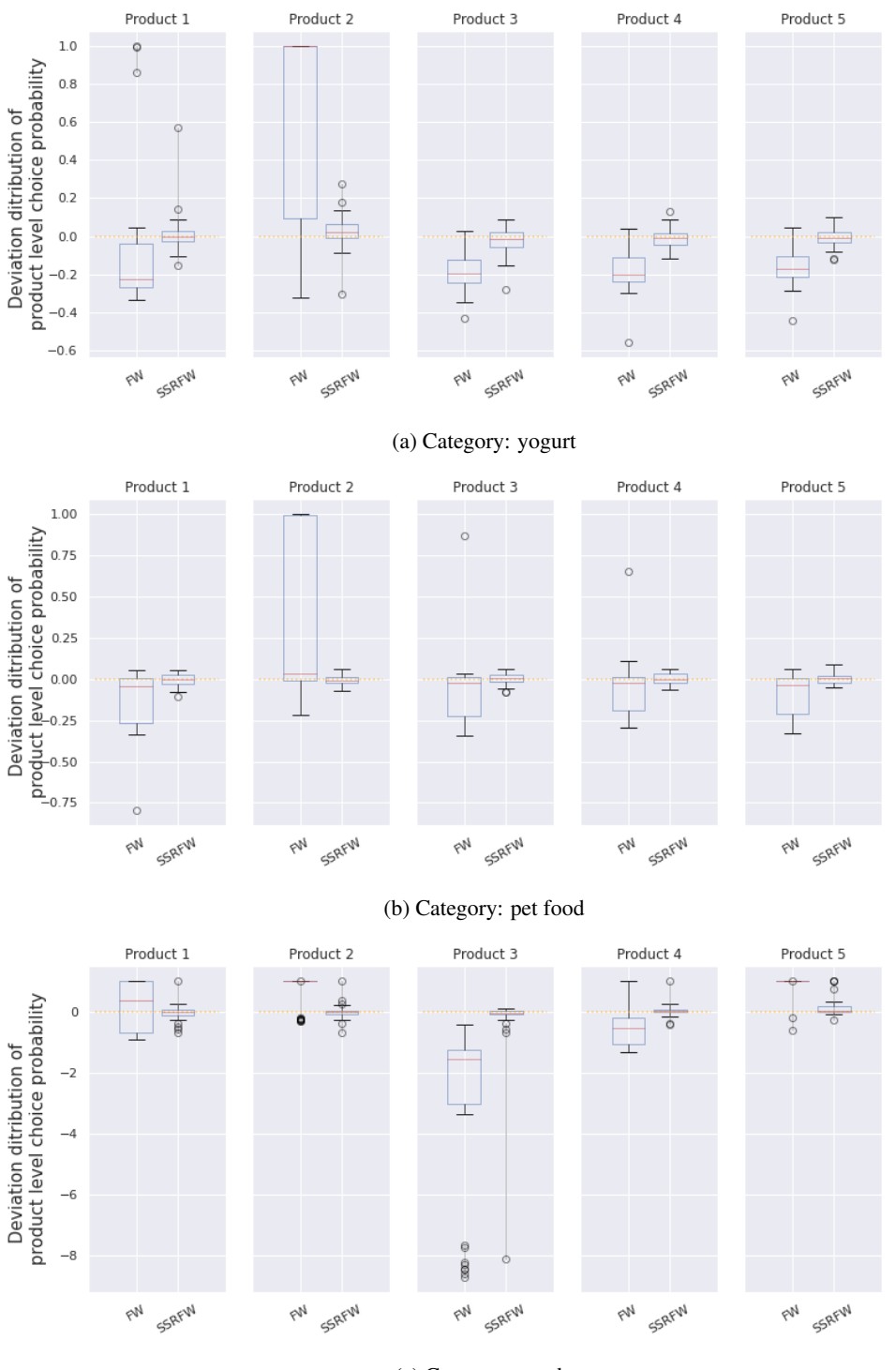

(a) Category: yogurt

(b) Category: pet food

(c) Category: candy

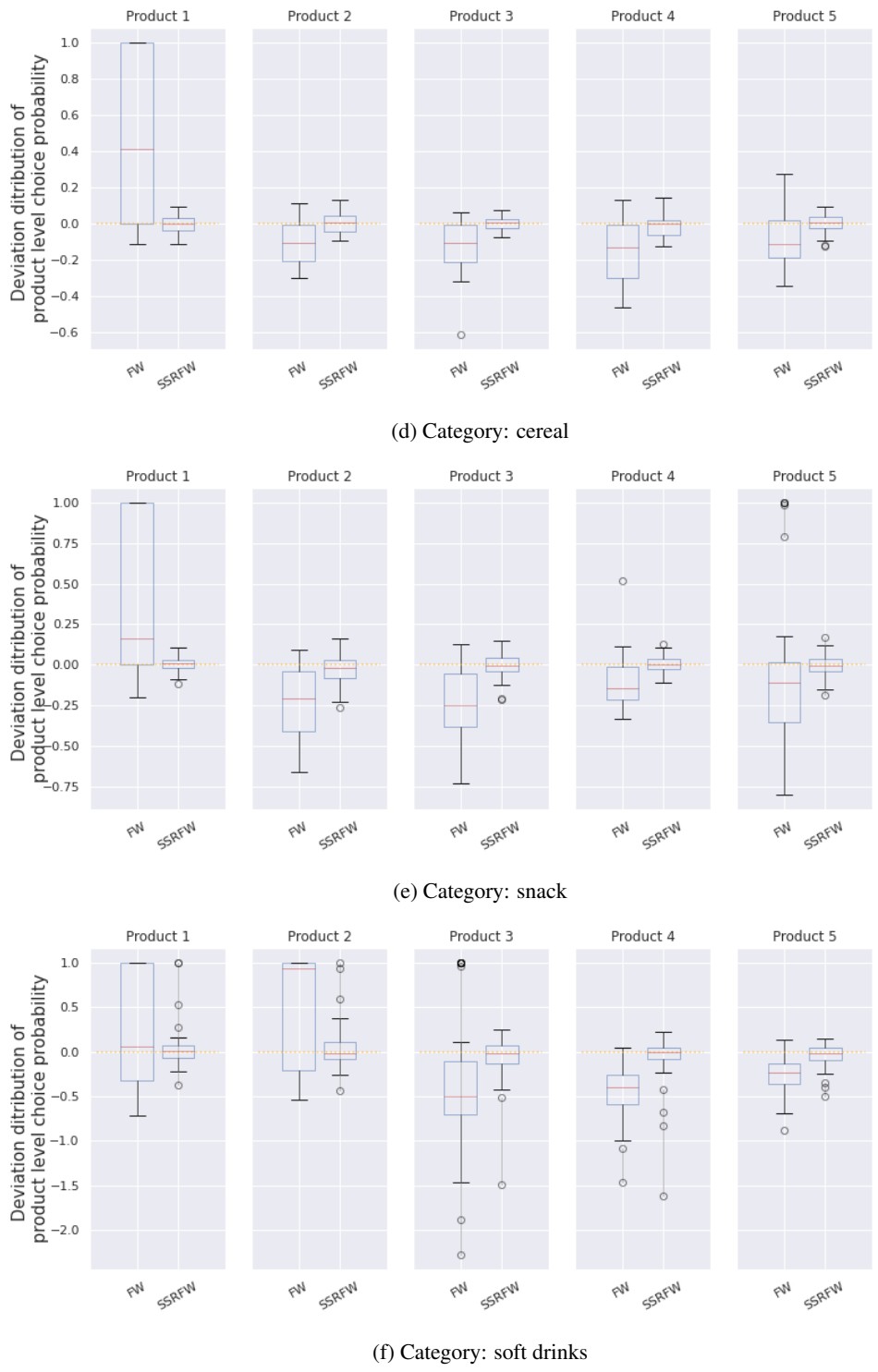

(d) Category: cereal

(e) Category: snack

(f) Category: soft drinks

Figure 8: Deviation-from-test distribution of product-level choice probability values

Next, we examine the number of iterations required until convergence. We also report the percentage of active directions for the given number of iterations. We think these two metrics measures the effectiveness of each direction being chosen during the learning process.

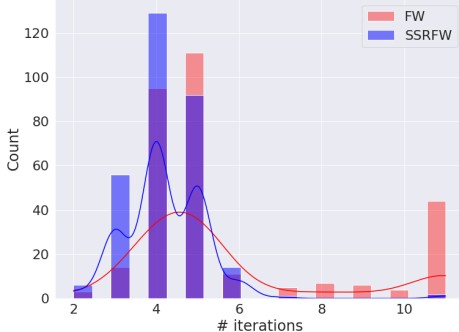
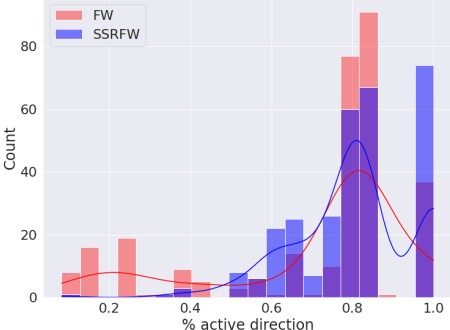

Figure 9: # iterations and % active directions

From Figure 9, we can see the FW algorithm usually requires more iterations, yet the percentage of active directions is lower than `SSRFW`. This suggests that more computation is wasted during the learning process.

The last metric we look at is the percentage of *boundary types* in the learning result. Figure 10 shows that the original FW still exhibits the same problem of generating boundary-type logit vectors while `SSRFW` is much less likely to suffer from this problem.

Figure 10: Percentage of boundary types

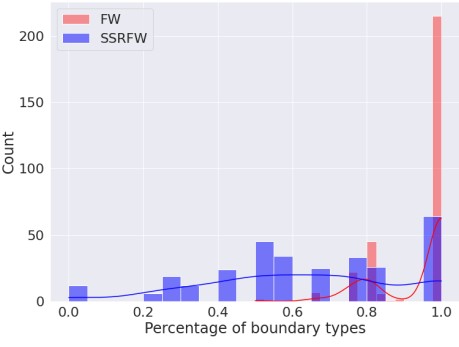

Finally, as requested by Kilts Center for Marketing at the University of Chicago School of Business, we make the following disclaimers:

- Researcher(s)' own analyses calculated (or derived) based in part on data from Nielsen Consumer LLC and marketing databases provided through the NielsenIQ Datasets at the Kilts Center for Marketing Data Center at The University of Chicago Booth School of Business.

- The conclusions drawn from the NielsenIQ data are those of the researcher(s) and do not reflect the views of NielsenIQ. NielsenIQ is not responsible for, had no role in, and was not involved in analyzing and preparing the results reported herein.

## C  Multi-Product Pricing Formulation

We provide a concrete example formulation for the *multi-product pricing* problem, where The objective is to maximize the total revenue by finding the optimal price for a set of $M$ products:

$$\max_{\boldsymbol{p}} \sum_{j=1}^{M} p_j \sum_{k=1}^{K} \alpha_k \frac{\exp \sigma(\mathbf{z}_j, \boldsymbol{p}; \boldsymbol{\beta}_k)}{1 + \sum_i \exp \sigma(\mathbf{z}_j, \boldsymbol{p}; \boldsymbol{\beta}_k)}$$

$$\text{s.t.} \quad p_j \geq 0 \qquad \forall j \in [M]$$

Note that the decision variables $\boldsymbol{p} \in \mathbb{R}^M$ are also part of the utility function $\sigma$. This is due to the fact that in the multi-product pricing setting, prices — not only the price of a product itself, but also prices of other products in the same set — are often an important factor that will impact people's choice behavior. Because of this entanglement, it does not suffice if we only learn the aggregated choice probabilities of the population. Instead, we have to accurately estimate the parameters for each individual MNL mixture before we can solve this optimization problem.

In the above formulation, we also include an offset option, which allows the consumer to choose not to purchase anything from the set. The probability of the offset can be expressed as $\frac{1}{1 + \sum_i \exp \sigma(\mathbf{z}_j, \boldsymbol{p}; \boldsymbol{\beta}_k)}$.