# OpenReview forum: "Learning Mixed Multinomial Logits with Provable Guarantees"
_NeurIPS.cc/2022/Conference — NeurIPS 2022 Accept_

### Official Review · Reviewer_Ldhd · 2022-06-28

**Rating:** 6
**Confidence:** 3
**Soundness:** 3 good
**Presentation:** 2 fair
**Contribution:** 3 good

**Summary:**

The authors provide theoretical analysis of a new "Stochastic Subregion Frank-Wolfe Algorithm" for learning mixed multinomial logits. The authors provide provable guarantees showing a polynomial number of samples is sufficient to achieve the performance guarantees. Simulations are provided in the supplementary material.

**Questions:**

Below are several questions that may be worth a clarification:

* The results conveyed through Theorem 1 only depend on $\epsilon$ and $\delta$, and do not take into account other problem parameters such as $K$, $M$, etc. How do these parameters affect the sample complexity?
* Are there any information-theoretic impossibility results that can be used for comparison? For example, results that show a relation between feasible expected error compared to number of samples? Is the dependence on $\epsilon$ and $\delta$ already theoretically optimal, or is it possible that algorithms with better complexity exist?
* What is the computational complexity of the algorithms compared to previous algorithms such as MLE or previous algorithms that achieve sub-linear sample complexity for aggregate probabilities?
* In the supplementary material, several details are swept under the rug. For example, line 457 states that "enough sample"s are sufficient to give a performance guarantee by law of large numbers, but the whole discussion of that section relies on a precise characterization of the sample complexity --- can this be clarified? In other words, what is the precise sample complexity for the statement of line 457 to be true and does it affect the final sample complexity claims in the main text?
* Finally, in the simulations, although the authors claim that "EM achieves a marginally better performance" in certain scenarios in line 655, Figure 6-(b) does not suggest that and it appears the EM with suitably chosen $K$ has much better performance especially at larger epochs. While the authors clarify that EM "suffers from its instability", it does not appear to be a significant factor in this experiment. Can this be clarified?

**Limitations:**

Limitations are not thoroughly discussed in this submission. It would be beneficial to discuss limitations in more detail.

**Strengths And Weaknesses:**

Overall, the topic and results are interesting. The smoothness of reading is affected by some typos that can be improved through further polishing.

**Originality.** The authors provide a novel algorithm that can be considered a variation of the Franke-Wolfe algorithm with a component that takes into account a "feasible region". This concept is quite interesting. Using this concept, the hyperparameter $K$ is not required in the algorithm which makes it appealing compared to algorithms that require knowledge of $K$.

**Quality.** The paper overall is interesting. The algorithms presented make sense. See below sections for a few questions (see below Questions section) and several notational confusions (see below clarity).

**Clarity.** The clarity in the paper is mostly obstructed by typos and notational issues. Below are several points during my review:
* Clarity of notations and typos. For example:
    - defining "$\|\| \cdot \|\|$" before it appears in Theorem 1.
    - in the description of Theorem 1, the summation should be over $k'$ and not $k$.
    - there are multiple typos throughout the submission but minimally affects the overall read. It would be beneficial to fix these typos in further revisions.

**Significance.** The MMNL model, Frank-Wolfe algorithm and related topics benefit a range of real world applications and therefore related research, including this paper, is important.

---

> ### Author Response · Authors · 2022-08-02
> **Response to Reviewer Ldhd Part II**
>
> * Finally, in the simulations, although the authors claim that "EM achieves a marginally better performance" in certain scenarios in line 655, Figure 6-(b) does not suggest that and it appears the EM with suitably chosen has much better performance especially at larger epochs. While the authors clarify that EM "suffers from its instability", it does not appear to be a significant factor in this experiment. Can this be clarified?
>
> Due to the problem scale (the model parameters and attributes are all randomly generated within the range of [-1, 1] while in reality this range is usually much larger), we think it is a better idea to look at the relative performance comparison instead of the absolute values. For instance, when EM suffers from the instability, the deviation of the estimators from the true values can be 100%-200% larger than that of our algorithm, which has much smaller fluctuations.
>
> On the other hand, the two information AIC and BIC both suggest we choose $K=3$ for the EM algorithm, which consistently have worse performance. In real world scenarios where we do not have the knowledge of the ground truth $K$, there is no guarantee which $K$ we will end up selecting and how far away it is from the true value. Such uncertainty and instability is the major motivation why we developed this algorithm so that we can minimize the risk of model misspecification as well as provide some theoretical guarantees on the estimators. We believe this comprehensive objective is more important to us and outweighs the task to only seek or develop learning algorithms/heuristics that can beat EM by a large margin.
>
> * References:
>
>
> F. Chierichetti, R. Kumar, and A. Tomkins. Learning a mixture of two multinomial logits. In Proceedings of the 35th International Conference on Machine Learning, volume 80 of Proceedings of Machine Learning Research, pages 961–969. PMLR, 10–15 Jul 2018.
>
> W. Tang. Learning an arbitrary mixture of two multinomial logits, 2020.
>
> Z. Zhao and L. Xia. Learning mixtures of plackett-luce models from structured partial orders. Advances in Neural Information Processing Systems, 32, 2019.

---

> > ### Comment · Reviewer_Ldhd · 2022-08-08
> > **Update after author response**
> >
> > I would like to thank the authors for their detailed responses. I am raising my score from 5 to 6.
> >
> > The author response above gives intuition of how $M$ would affect sample complexity --- a clarification of this dependency in the main results, instead of using a big-O notation and ignoring the dependence, would be helpful.
> >
> > The questions about comparisons against EM and clarifications of supplementary material were well addressed.

---

> > > ### Author Response · Authors · 2022-08-09
> > > **Thanks to Reviewer Ldhd's response**
> > >
> > > We thank the reviewer for his/her time to read through our response. We have also updated the submission to include a more detailed discussion on the impact of $M$ to the sample complexity at the end of Appendix A.2. Please let us know if you have any further questions.

---

> ### Author Response · Authors · 2022-08-02
> **Response to Reviewer Ldhd Part I**
>
> We thank the reviewer for his/her time and detailed comments. Below we will address the questions from the reviewer and make additional clarifications if needed.
>
> * Clarity of notations and typos. For example:
>
>   - defining $\lVert \cdot \rVert$ before it appears in Theorem 1.
>   - in the description of Theorem 1, the summation should be over $k'$ and not $k$
>   - there are multiple typos throughout the submission but minimally affects the overall read. It would be beneficial to fix these typos in further revisions.
>
> We appreciate the reviewer's attention to the details and pointing out the typos. For the first point, we use the standard Euclidean norm and we will make sure to address these issues and/or add explicit definition in the updated version.
>
>
>
>
> * The results conveyed through Theorem 1 only depend on $\epsilon$ and $\delta$, and do not take into account other problem parameters such as $K$, $M$, etc. How do these parameters affect the sample complexity?
>
> Intuitively, we think $M$ has some implicit impact on the sample complexity. In particular, if we have an extremely large $M$ while only collect the data for a small $T$ periods, then the empirical choice distribution over the $M$ choices will be very sparse, making the score function computed based on the empirical CDF less credible. One thing worth noting is that in many real world applications where the choice set is usually a substitution set, $M$ will in general be a small number.
>
> The sample complexity does not depend on $K$ or $L$ as the data we collect are the choice data from all the decision makers in the population among a fixed set of items. However, $K$ and $L$ do have an impact on the computational complexity. More specifically, we need to repeat the subsample construction process for $L$ times, therefore the larger $L$ is, the more computational cost there will be. $K$ has an implicit impact on $L$ in that a larger $K$ usually means we need to create more subsamples to ensure that we can cover every single mixture type.
>
>
> * Are there any information-theoretic impossibility results that can be used for comparison? For example, results that show a relation between feasible expected error compared to number of samples? Is the dependence on $\epsilon$ and $\delta$ already theoretically optimal, or is it possible that algorithms with better complexity exist?
>
> We think this is a great future research question to explore in this field. The study on mixed multinomial logit model is still in very early stage, and research work that aims to exploit the structure of the model to derive or prove theoretical properties have been limited to 2-MNL models only; what we found include Chierichetti et al. [2018], Zhao et al. [2019], Tang [2020].
>
> * What is the computational complexity of the algorithms compared to previous algorithms such as MLE or previous algorithms that achieve sub-linear sample complexity for aggregate probabilities?
>
> The SSRFW algorithm inherits the original FW framework with the major difference in that, so the sub-linear convergence of the aggregated probabilities still applies in our approach. We do have some additional computational cost resulted from the $\mathcal Q$ construction process, with the influences of hyperparameters such as $L$ and $K$ discussed above.
>
>
> * In the supplementary material, several details are swept under the rug. For example, line 457 states that "enough sample"s are sufficient to give a performance guarantee by law of large numbers, but the whole discussion of that section relies on a precise characterization of the sample complexity --- can this be clarified? In other words, what is the precise sample complexity for the statement of line 457 to be true and does it affect the final sample complexity claims in the main text?
>
> In the supplementary material, Appendix A is mainly about the proof of the main theorem. It is organized in the following way: we first demonstrate that with/assuming  we have enough samples, we can achieve the convergence result (in A.1). Then we discuss what we mean by "enough samples" in A.2, which helps us quantify the sample complexity. These two parts combined together finally allow us to prove the main result (in A.4).
>
> The precise sample complexity $T$, in particular, is determined by bounding the score function $s(i,j)$, defined on line 202, such that it is smaller than $\epsilon$ when $i$ and $j$ are of the same type and larger than $1-\epsilon$ when they are of different types. This allows us to relate the sample complexity, i.e. the number of time periods we need to collect the data for, to the tolerance $\epsilon$, as shown in A.2.1 and A.2.2.

---

### Official Review · Reviewer_oo6h · 2022-07-10

**Rating:** 6
**Confidence:** 2
**Soundness:** 3 good
**Presentation:** 3 good
**Contribution:** 2 fair

**Summary:**

The paper proposed a new method for learning mixtures of multinomial logits based on historical choice data. The proposed method is based on Frank-Wolfe while restricting to a narrowed set of estimated choice probability vectors. The paper provided theoretical results on convergence and sample complexity. The paper provided numerical experiments to demonstrate the performance of the proposed method.


**Questions:**

- Line 34 : "But it does not guarantee the estimators’ convergence." should better be "But it does not guarantee the estimators’ convergence to the true parameters", because EM algorithms do converge to something.

- The authors could clarify why the problem considered in the paper is important and particularly relevant to the ML community.

- The authors may provide more details on the underlying model for the historical choice data. Specifically, how were the parameters $z_j$'s (line 51) connected to the decision maker's choice probability q_k? Also, what's the usual relationship between N and T？ Do we consider a large N regime, large T regime, or large N large T regime?

- Theorem 1: what is the sample dependency on K (or L) and M? Also, is there a convergence guarantee for $\beta$'s in addition to q and \alpha?

- The method doesn't need to choose K but needs to choose an additional parameter L. How do you choose L in practice?

- Algorithm 1 step 3 : $v$ was not defined.

- Algorithm 1 : what's the stopping condition?

**Strengths And Weaknesses:**


Strengths. The proposed method seems effective for learning multinomial logit mixtures. The theoretical results are strong and solid. The paper is overall well-written.

Weaknesses. The paper specifically considers historical choice data, which should be reflected in the title. It seems a narrow topic of learning multinomial logit mixtures using historical choice data, which may reduce the paper's significance. The Preliminaries subsection is a bit brief and it would be more helpful to formally define the historical choice data.


- Line 35: typo of "Frank-Wolf (FW)"

- Line 330: typo of "Not only does it resolves ..."

---

> ### Author Response · Authors · 2022-08-02
> **Response to Reviewer oo6h Part II**
>
> * Theorem 1: what is the sample dependency on K (or L) and M? Also, is there a convergence guarantee for $\beta$'s in addition to q and $\alpha$?
>
> The sample complexity does not depend on $K$ or $L$ as the data we collect are the choice data from all the decision makers in the population among a fixed set of items. However, $K$ and $L$ do have an impact on the computational complexity. More specifically, we need to repeat the subsample construction process for $L$ times, therefore the larger $L$ is, the more computational cost there will be. $K$ has an implicit impact on $L$ in that a larger $K$ usually means we need to create more subsamples to ensure that we can cover every single mixture type.
>
> The learning of $\beta_k$'s is briefly discussed between line 233 and 242, where we can apply MLE after we have learned $\mathbf q_k$'s. Its convergence rooted from the property of maximum likelihood estimators under appropriated assumptions (e.g. identifiability of the model, compactness of the parameter space, etc.).
>
>
> * The method doesn't need to choose K but needs to choose an additional parameter L. How do you choose L in practice?
>
> We can use cross-validation to find a good value of $L$. For instance, we can use the training set to learn a set of parameters $\hat{\alpha}_k$'s and $\hat{\beta}_k$'s. We can then compute the distance between $\sum_k \hat{\alpha}_k q_k(\hat{\beta_k})$ and the aggregated choice probability in the test set. If the discrepancy is always large, that indicates we probably have missed one or more mixtures when we construct the $\mathcal Q$ set, hence we should try to increase the value of $L$.
>
>
> * Algorithm 1 step 3 : $v$ was not defined.
>
> $v$ is a dummy decision variable. Once we solve the linear subproblem on Line 3, its value will be assign to $q^{(k)}$ and becomes the new MNL logit vector generated at iteration $k$.
>
> * Algorithm 1 : what's the stopping condition?
>
> In general, we consider the following two stopping conditions: 1) the algorithm reaches the convergence of $\mathbf g$ to the ground truth value $\mathbf g^{*}$; 2) we reached a \texttt{max\_iter} value (this can help prevent overfitting).
>
>
> * typos:
>   - Line 35: typo of "Frank-Wolf (FW)"
>   - Line 330: typo of "Not only does it resolves ..."
>
> Thanks for pointing out the typos. We will correct these in the updated version.
>
> * References:
>
> F. Chierichetti, R. Kumar, and A. Tomkins. Learning a mixture of two multinomial logits. In Proceedings of the 35th International Conference on Machine Learning, volume 80 of Proceedings of Machine Learning Research, pages 961–969. PMLR, 10–15 Jul 2018.
>
> A. Seshadri, S. Ragain, and J. Ugander. Learning rich rankings. In H. Larochelle, M. Ranzato, R. Hadsell, M. Balcan, and H. Lin, editors, Advances in Neural Information Processing Systems, volume 33, pages 9435–9446, 2020.
>
> Z. Zhao and L. Xia. Learning mixtures of plackett-luce models from structured partial orders. Advances in Neural Information Processing Systems, 32, 2019.

---

> > ### Comment · Reviewer_oo6h · 2022-08-07
> > **Thank you for your response**
> >
> > Thank you for your response, which has adequately addressed my comments.

---

> > > ### Author Response · Authors · 2022-08-09
> > > **Thanks to Reviewer oo6h's response**
> > >
> > > We thank the reviewer for his/her time to read through our response. Please let us know if you have any further questions.

---

> ### Author Response · Authors · 2022-08-02
> **Response to Reviewer oo6h Part I**
>
> We thank the reviewer for his/her time and detailed comments. Below we will address the questions from the reviewer and make additional clarifications if needed.
>
> * The paper specifically considers historical choice data, which should be reflected in the title. It seems a narrow topic of learning multinomial logit mixtures using historical choice data, which may reduce the paper's significance. The Preliminaries subsection is a bit brief and it would be more helpful to formally define the historical choice data.
>
>   The authors could clarify why the problem considered in the paper is important and particularly relevant to the ML community.
>
> We believe using specific types of data to analyze people's choice behavior has been widely adopted in this research field. For instance, Chierichetti et al. used the choice distribution data to learn 2-MNL models [ICML 2018], and Zhao et al. and Seshadri et al. used ranking data to learn mixed Plackett-Luce and mixed MNL models [NeurIPS 2019, NeurIPS 2020]. In addition, we think our algorithm and this research topic in general can have a big impact and benefit many downstream applications. While we all agree that developing good machine learning algorithms is one of the central topics in our community, we believe it is equally important to understand how to best utilize the learning outcome and incorporate it into decision making in real world problems. For instance, there are many e-commerce providers that have accumulated rich choice data from consumer interactions and transactions online. If they can develop some trustworthy models to understand consumers' choice preferences, there are many operational strategies they can take to further boost the business, such as identifying features that consumers like for future product development, offer better personalization services, e.g. personalized recommendation, or build optimal pricing models to maximize revenue, just to name a few. (We provide a concrete application problem formulation in [this comment](https://openreview.net/forum?id=NoAZRVthZL&noteId=F4ZGmCG3Fu).)
>
> We used the common definition of choice data and introduced the data setting in Section 1 and 2 as the following: "for each time period $t=1,\dotsc, T$, we assume each decision maker $i$'s choice $X_i^{(t)}$ are i.i.d categorical random variables with support $x \in [M]$ and probability mass function (PMF) specified by $\mathbf q_k$ if he/she is of type $k$." In other words, a decision maker $i$ will have data $x_i^{(1)}, x_i^{(2)}, \dotsc, x_i^{(T)}$ recording his/her choices among the $M$ choices. We then compute the aggregated choice probability $\mathbf y^{t} = \frac{1}{N} \sum_{i=1}^{N} Y_i^{t} \in \mathbb R ^{M}$ where we introduced a dummy random variable $Y_i^{t}=[0,\dotsm,1,\dotsm,0]^{\top}\in \mathbb R^M$, where $Y_{ij}^{t}=1$ if $x_{i}^{(t)}=j$.
>
>
> * Line 34 : "But it does not guarantee the estimators’ convergence." should better be "But it does not guarantee the estimators’ convergence to the true parameters", because EM algorithms do converge to something.
>
> Thanks for the suggestion. We agree that we should be more explicit as the EM algorithm will at least converge to a local optima. We will correct this statement in the updated version.
>
>
>
> * The authors may provide more details on the underlying model for the historical choice data. Specifically, how were the parameters $z_j$'s(line 51) connected to the decision maker's choice probability $q_k$? Also, what's the usual relationship between $N$ and $T$? Do we consider a large $N$ regime, large $T$ regime, or large N large T regime?
>
>
> We'd like to start by clarify that $z_j$'s are not parameters. They are observed attributes of the options in the choice set. Examples of attributes can be whether it is organic if we think about food items, or whether it is public transportation when we model people's commute options. Under linear utility assumption, the probability of choosing $j$-th item for a particular mixture $k$ is
> $$q_j(\beta_k) = \frac{\exp(\beta_k^{\top} z_j)}{\sum_{i=1}^M \exp(\beta_k^{\top} z_i)}  \quad \text{ (line 59)}$$
>
> In general, $N$ is the number of decision makers in the population, which can be large or small depending on the application. However, we usually assume that $N \gg K$. On the other hand, $T$ is the number of repetitive choices that have been made. The larger $T$ is, the more accurate our estimates will be; yet as we have shown in the simulation studies, we believe a moderate $T$ around 50 can already give us relatively good estimates.

---

### Official Review · Reviewer_VDBV · 2022-07-12

**Rating:** 7
**Confidence:** 3
**Soundness:** 3 good
**Presentation:** 3 good
**Contribution:** 3 good

**Summary:**

The authors provide theoretical results on learning mixed multinomial logits. The main contribution is a new algorithm that learns both mixture weights and component-specific logit parameters with provable convergence guarantees for an arbitrary number
of mixtures.

**Questions:**

See the main comments.

**Strengths And Weaknesses:**

[Strengths]
1. The paper is clearly written and the theoretical results look solid.
2. Sample complexity is established for the proposed algorithm.

[Weaknesses]
1. The proposed algorithm performs EM-like updates. While sample complexity is provided, the algorithm could take many iterations to converge, and potentially converge to a local optima. How does the algorithm address these issues?
2. Both \alpha and \beta depend on the parameter \epsilon in Definition 1. Since the magnitude of z could be arbitrary, is the condition on \beta necessary?
3. I think the authors forgot to include the simulation studies as suggested in the abstract.

---

> ### Author Response · Authors · 2022-08-02
> **Response to Reviewer VDBV**
>
> We thank the reviewer for his/her time and detailed comments. Below we will address the questions from the reviewer and make additional clarifications if needed.
>
> * The proposed algorithm performs EM-like updates. While sample complexity is provided, the algorithm could take many iterations to converge, and potentially converge to a local optima. How does the algorithm address these issues?
>
> We would like to clarify that our algorithm has several critical differences from the EM algorithm. First, the Frank-Wolfe framework will generate the aggregated estimator $\mathbf g$ that *converges* to global optima, which is a property of Frank-Wolfe itself. This property is stated in Lemma 2 in Appendix A in the supplementary material. Second, we use historical choice data to construct the extreme point set for the Frank-Wolfe framework, which, combined with the first point, enables the convergence of individual component level parameters. Both are not achievable in the EM algorithm, which is one of the key motivations that we developed the SSRFW algorithm to eliminate the instability (in convergence) of the EM and provide provable guarantees on the estimators.
>
> In addition, despite being an iterative algorithm, EM performs alternating minimization steps after they have pre-determined the number of mixtures $K$ --- each iteration then updates the parameter values to make improvements. In comparison, our algorithm identifies a new direction, corresponding to a mixture component in each iteration, hence reducing the chance of model misspecification from inputting a wrong value of $K$. During this process, the new direction is chosen by solving a linear subproblem in Line 3 in Algorithm 1, therefore the maximum number of iterations that can potentially happen is determined by $L$, which according to Theorem 4, is impacted by the value $\alpha_{\text{min}}$. However, in the experiments we never observed that our algorithm will ever reach up to $L$ iterations. It usually stops after a few more than the ground-truth $K$ values.
>
>
> * Both $\alpha$ and $\beta$ depend on the parameter $\epsilon$ in Definition 1. Since the magnitude of $z$ could be arbitrary, is the condition on $\beta$ necessary?
>
> While we adapted the notations from Kalai et al. [2010], we realize this can cause some confusions. The $\epsilon$ in Definition 1 does not necessarily have to be the same as the tolerance $\epsilon$ in Theorem 1. In Definition 1, $\epsilon$ regulates the parameters (e.g. boundedness of $\beta$) and the separation between different mixture components (so there do not exist two identical MNL components in the mixture model). In Theorem 1, $\epsilon$ is the tolerance level for the deviation between estimators and true values. While it is true both are used to refer to an arbitrary small number, they are indeed quantifying two independent metrics. We appreciate the reviewer's attention to detail and we will make this point clear in the updated version.
>
> * I think the authors forgot to include the simulation studies as suggested in the abstract.
>
> We apologize for the confusion. Due to page limit, the entire numerical experiments, including simulation and case studies have been moved to Appendix B in the supplementary material. We are happy to provide more responses, should the reviewer have additional questions after reading the content.

---

### Official Review · Reviewer_zoBe · 2022-07-12

**Rating:** 6
**Confidence:** 3
**Soundness:** 2 fair
**Presentation:** 2 fair
**Contribution:** 3 good

**Summary:**

This paper studies the recovery of mixed multinomial logits using Frank-Wolfe algorithms. The authors proved that the proposed algorithm SSRFW converges to the true parameter of each individual mixture model. They also provided a sample complexity analysis of the proposed algorithm.


**Questions:**

Regarding the convergence of Algorithm 1 and Algorithm 2, I wonder whether the authors considered the computational complexity of Algorithm 2 in the analysis of the complexity of SSRFW. Specifically, how many iterations do step 4 of Algorithm 2 take to converge? And what is the order of $L$ in Algorithm 2 in order to get a good estimate of the input set $\mathcal{Q}$?

Can the authors elaborate more on the reason for obtaining the individual mixture? For constrained optimization with an aggregated objective, isn’t it enough to just know the aggregate prediction?

What is $\alpha$ in Line 4 of Algorithm 1? It does not appear in the objective function. How are you going to minimize $\mathcal{L}$ over it? Moreover, the footnote says $\alpha$ is a $k+1$ dimensional vector, while in Line 171 of the paper, the authors say $\alpha^{(k)}=\alpha$ is a $k$-dimensional vector (by Line 4 of Algorithm 1). This inconsistency makes it hard to understand the algorithm.


**Limitations:**

For the experiment results, I think it would be more reasonable to also compare the proposed method with those in recent related papers such as Chierichetti et al. [2018] and Jagabathula et al. [2020]. Though these baseline algorithms may work in a reduced setting or have different theoretical guarantees, their methods should still be applicable to the empirical studies in this paper. Therefore, it would be interesting to see how they compare with one another in the experiments.

Lin 336: “reply” => “rely”


**Strengths And Weaknesses:**

The paper is generally well written and the results are interesting. Some clarification on the sample complexity of the subroutine should be added to the analysis. The comparison in the experiments could also be improved. See my specific questions in the next section.

---

> ### Author Response · Authors · 2022-08-02
> **Response to Reviewer zoBe Part II**
>
> * What is $\alpha$ in Line 4 of Algorithm 1? It does not appear in the objective function. How are you going to minimize $\mathcal L$ over it? Moreover, the footnote says $\alpha$ is a $k+1$ dimensional vector, while in Line 171 of the paper, the authors say $\alpha^{(k)}=\alpha$ is a $k$-dimensional vector (by Line 4 of Algorithm 1). This inconsistency makes it hard to understand the algorithm.
>
> We apologize for the typo on Line 171: the correct sentence should go as ``$\alpha^{(k)} \in \mathbb R^{k+1}$ represents the mixture weight vector for the $k+1$ choice probability vectors we have generated so far...''. We appreciate the reviewer's attention to detail and we will correct this typo in the updated version. The notation in the FW framework is rather complicated and we would like to use this opportunity to provide some further explanations.
>
>
> $\alpha^{(k)}$ in Algorithm 1 is a growing size vector, which changes its size as the number of iteration increases. At each iteration, we generate a new direction $q^{(k)}$, which is viewed as the choice probability vector associated with the mixture $k$. More specifically, at iteration $k$ and after line 3 finishes, we have obtained $q^{(0)}, q^{(1)}, \dotsc, q^{(k)}$. Then we want to find the best mixture weights to be distributed among with these $k+1$ vectors by minimizing $\mathcal L$, namely
>
>
> $$\min_{\alpha^{(k)}\in \Delta_k} \frac{1}{2} \sum_{t=1}^{T} \lVert \sum_{s=0}^{k} \alpha^{(k)}_s q^{(s)} - y^{t}\rVert^{2}$$
>
> where $\alpha^{(k)}_s$ refers to the $s$-th element in the vector $\alpha^{(k)}$, corresponding to the mixture associated with $q^{(s)}$. The decision variables in this step are the $k+1$ elements in the $\alpha^{(k)}$ vector. This is a standard (constrained) concave optimization, which can be solved very efficiently.
>
> * For the experiment results, I think it would be more reasonable to also compare the proposed method with those in recent related papers such as Chierichetti et al. [2018] and Jagabathula et al. [2020]. Though these baseline algorithms may work in a reduced setting or have different theoretical guarantees, their methods should still be applicable to the empirical studies in this paper. Therefore, it would be interesting to see how they compare with one another in the experiments.
>
> We would like to clarify that the numerical experiments we ran in the appendix mainly focused on the comparison between the our framework (SSRFW) and the original FW by Jagabathula et al. [2020]. On the other hand, the algorithm for uniform 2-MNL model developed in Chierichetti et al. [2018] has very different assumptions on the data, hence we found it hard to conduct a fair comparison. More specifically, they assume there exists an oracle that will return the ground-truth aggregated choice probability for any arbitrary subset $S \subseteq \\{1,\dotsc,M\\}$, where we assumed our data are based on repetitive choice data of the set $\\{1,\dotsc,M\\}$.
>
> * Line 336: "reply" => "rely"
>
> Thanks for pointing out the typo. We will correct it in the updated version.

---

> > ### Comment · Reviewer_zoBe · 2022-08-09
> > **Thanks for the detailed response**
> >
> > Thanks for your detailed response. I have raised my score for this submission.

---

> > > ### Author Response · Authors · 2022-08-09
> > > **Thanks to Reviewer zoBe's response**
> > >
> > > We thank the reviewer for his/her time to read through our response. Please let us know if you have any further questions.

---

> ### Author Response · Authors · 2022-08-02
> **Response to Reviewer zoBe Part I**
>
> We thank the reviewer for his/her time and detailed comments. Below we will try to address the questions from the reviewer and make additional clarifications if needed.
>
> * Regarding the convergence of Algorithm 1 and Algorithm 2, I wonder whether the authors considered the computational complexity of Algorithm 2 in the analysis of the complexity of SSRFW. Specifically, how many iterations do step 4 of Algorithm 2 take to converge? And what is the order of $L$ in Algorithm 2 in order to get a good estimate of the input set $\mathcal Q$?
>
> We believe the number is mainly guided by the quantity $\frac{n}{\alpha_k}$ and we provide an intuitive explanation. Recall that $\alpha_k$ is the proportion of the decision maker of type $k$ in the population. Assume we are trying to create a subsample originated from decision maker $i$, who is of type $k$. In an extreme case where after we sampled a decision maker $j$ from the population, we accept $j$ into the subsample with probability 1 if $j$ is in-type, and we reject $j$ also with probability 1 if $j$ is out-type. Denote $T$ as the total number of iterations in the while loop of Algorithm 2; essentially $T$ represents the number of samples we try to draw from the population. If we sample from the population $\frac{n}{\alpha_k}$ times, we expect to have $\frac{n}{\alpha_k} \cdot \alpha_k =n$ in-type decision makers being sampled, all of which will be accepted and giving us the desired subsample of size $n$. In the actual case, though not precisely at 1, we have designed the accepting probability $p_{j|i}$ to be at least $1-\epsilon$ for in-type and at most $\epsilon$ for out-type (given appropriate number of samples), such that we achieve the almost same situation as in the previous scenario.
>
> In Theorem 4, we provide a bound for the expected number of $L$, which is $\frac{1}{\alpha_1}\log \frac{1}{\alpha_1}$, assuming $\alpha_1 \le \dotsc \le \alpha_K$. The intuition is that if the minimum mixture weight is very small, then inevitably we need to create more subsamples (i.e. a larger $L$) to make sure all mixture type is being chosen at least once in constructing an element in the set $\mathcal Q$.
>
> * Can the authors elaborate more on the reason for obtaining the individual mixture? For constrained optimization with an aggregated objective, isn’t it enough to just know the aggregate prediction?
>
> There are many important applications in operations research that can benefit from knowing the individual mixture choice preference, from minimizing the risk of model misspecification to offering personalized service such as product recommendation. In addition, it helps to predict the system dynamic and offer a higher flexibility in modeling a time-varying environment. For instance, if we change the attributes of the options over time, we can still easily compute the choice probability upon such change if we have learned the individual mixture weights $\alpha_k$ and preference vector $\beta_k$, while the aggregated choice probability can only be applied in a limited number of settings.
>
> We would also like to provide a concrete example in one of the most important applications in operations research: pricing, where our objective is to learn the optimal price for a set of products simultaneously to maximize the revenue. The problem is formulated as follows:
> \begin{align*}
> 	& \max_{\mathbf p} ~ \sum_{j=1}^M p_j \sum_{k=1}^K \alpha_k \frac{\exp{\sigma(\mathbf z_j, \mathbf{p};\beta_k)}}{1+\sum_i \exp{\sigma(\mathbf z_j, \mathbf{p}; \beta_k)}} \\\\
> 	\text{s.t.}\quad & p_j \ge 0 \qquad \forall j \in [M]
> \end{align*}
>
> where $\sigma$ is the utility function parameterized by $\beta_k$ and takes the product attributes as input (the $+1$ in the denominator comes from introducing an offset option that allows the consumer to choose to not purchase anything from the set). Often, the price of the products has a large impact on consumers' decisions and needs to be thought as part of the product attributes. We can see from the problem formulation that when $\mathbf p$ changes, the individual choice probabilities as well as the aggregated choice probability will all change simultaneously. In other words, if we learn the aggregate prediction and fix it, we will not be able to carry out many downstream applications such as this one which depends on the individual MNL parameters.

---

### Author Response · Authors · 2022-08-09
**Thanks to all the reviewers and updated submission**

We would like to thank all the reviewers for their detailed comments and engagement in the discussion, which has been of great help to improve the quality of our paper.

We have updated our submission, mainly to reflect the following changes:
* Fixed multiple typos as suggested by the reviewers
* Added missing definitions and reasonings where confusions can happen
* Included more discussion on the sample complexity at the end of Appendix A.2

Last but no least, we extended our case study to six category of products and included a much more extensive analysis on the results (p23-27).

Should the reviewers have any more questions, please let us know and we are happy to answer. Thanks again to all the reviewers and AC for their time and consideration.

---

### Meta-Review · Area_Chair_FUwv · 2022-08-21

**Recommendation:** Accept
**Confidence:** Certain

**Metareview:**

This paper proposes Frank-wolfe method for learning mixture of multinomial logit models. The algorithms is novel and the first of its kind to be applied to this problem. This is an important addition to the widely studied topic of preference learning, and this result extends the capability of what can be efficiently learned under realistic scenario with mixture models. Mixture models for ranking are notoriously hard problems and a novel approach such as the one introduced in this paper is critical in bridging the gap to making preference learning practical. All the reviewers' concerns have been addressed in the rebuttal adequately.

Given the good quality of the paper, it will be quite beneficial for the readers if the paper spend some more time on explaining how the result compares to the extensive work that has been don on mixture of multinomial logit models. Such a nicely written related work section will only make the paper more interesting and broaden the impact of the results. Some important and closely related work that predate [Chierichetti et al. 2018] are missing. Some more recent related work are missing. Here is a sampling of such related work that should be compared with:
- Learning Mixtures of Random Utility Models with Features from Incomplete Preferences Zhibing Zhao, Ao Liu, Lirong Xia, (IJCAI-22)
- On the identifiability of mixtures of ranking models, Xiaomin Zhang, Xucheng Zhang, Po-Ling Loh, Yingyu Liang, https://arxiv.org/abs/2201.13132
- Learning Mixtures of Plackett-Luce Models from Structured Partial Orders, Zhibing Zhao, Lirong Xia, (NeurIPS 2019)
- Learning Plackett-Luce Mixtures from Partial Preferences, Ao Liu, Zhibing Zhao, Chao Liao, Pinyan Lu, Lirong Xia (AAAI-19)
- Learning Mixtures of Plackett-Luce Models, Zhibing Zhao, Peter Piech, Lirong Xia, (ICML-16)
- Collaboratively Learning Preferences from Ordinal Data, Sewoong Oh, Kiran K. Thekumparampil, Jiaming Xu, (NIPS 2015)
- A Topic Modeling Approach to Ranking, Weicong Ding, Prakash Ishwar, Venkatesh Saligrama, (AISTATS-15)
- Learning mixed multinomial logit model from ordinal data, S Oh, D Shah (NIPS-14)






**Award:**

No

---

### Decision · Program_Chairs · 2022-09-14

Accept